# FouriDown: Factoring Down-Sampling into Shuffling and Superposing

Qi Zhu [1]*, Man Zhou [2,3]*, Jie Huang [1], Naishan Zheng [1], Hongzhi Gao [1],
Chongyi Li [4], Yuan Xv [3], Feng Zhao [1]†

[1]University of Science and Technology of China,
[2]S-Lab, [3]Nanyang Technological University, [4]Nankai University
{zqcrafts,hj0117,nszheng, hongzhigao}@mail.ustc.edu.cn,
{man.zhou, xu.yuan}@ntu.edu.sg, lichongyi25@gmail.com, fzhao956@ustc.edu.cn

## Abstract

Spatial down-sampling techniques, such as strided convolution, Gaussian, and Nearest down-sampling, are essential in deep neural networks. In this paper, we revisit the working mechanism of the spatial down-sampling family and analyze the biased effects caused by the static weighting strategy employed in previous approaches. To overcome the bias limitation, we propose a novel down-sampling paradigm in the Fourier domain, abbreviated as FouriDown, which unifies existing down-sampling techniques. Drawing inspiration from the signal sampling theorem, we parameterize the non-parameter static weighting down-sampling operator as a learnable and context-adaptive operator within a unified Fourier function. Specifically, we organize the corresponding frequency positions of the 2D plane in a physically-closed manner within a single channel dimension. We then perform point-wise channel shuffling based on an indicator that determines whether a channel's signal frequency bin is susceptible to aliasing, ensuring the consistency of the weighting parameter learning. FouriDown, as a general operator, comprises four key components: 2D discrete Fourier transform, context shuffling rules, Fourier weighting-adaptively superposing rules, and 2D inverse Fourier transform. These components can be easily integrated into existing image restoration networks. To demonstrate the efficacy of FouriDown, we conduct extensive experiments on image de-blurring and low-light image enhancement. The results consistently show that FouriDown can provide significant performance improvements. The code is publicly available to facilitate further exploration and application of FouriDown at *https://github.com/zqcrafts/FouriDown*.

## 1 Introduction

Down-sampling technique [1, 2, 3] plays a vital role in deep neural networks because of its benefits in enlarging the receptive field, extracting hierarchical features, improving computational efficiency, and handling scale and translation variations. However, based on the signal sampling theorem, existing down-sampling techniques such as strided convolution, Gaussian, and Nearest down-sampling [1, 4, 5, 6] unavoidably reduce the sampling frequency of discrete signals, leading to unexpected frequency aliasing where high frequencies are folded into low frequencies.

To address the aliasing problem, several strategies [7, 8, 9, 10, 11, 12, 13] have been developed. They pre-process the signals applying the low-pass filtering mechanism, which aims to filter out high-frequency information by employing different types of low-pass designs. There are two commonly

---

*Both authors contributed equally to this research.
†Corresponding author.

37th Conference on Neural Information Processing Systems (NeurIPS 2023).

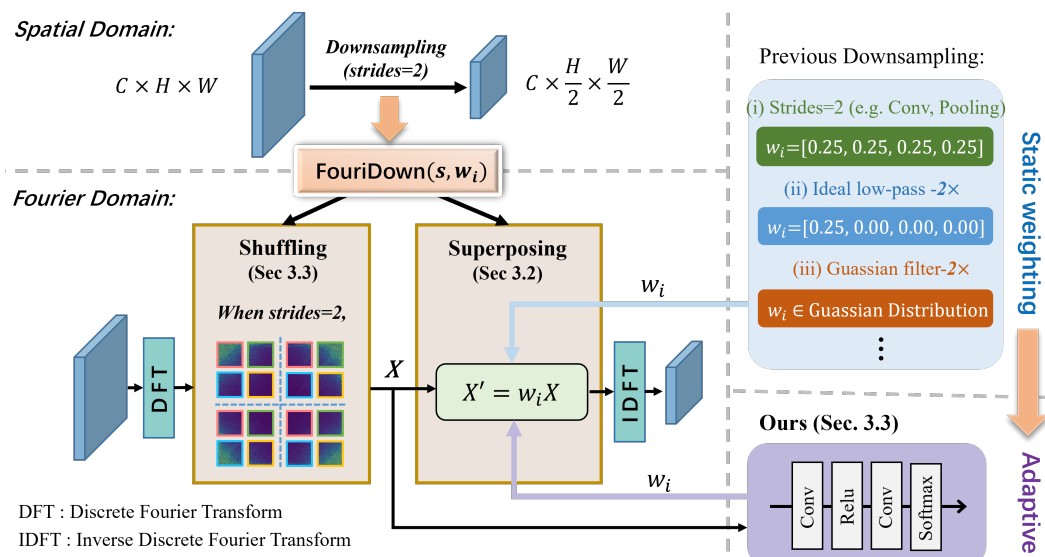

Figure 1: Comparison on the flowcharts of different down-sampling techniques in $2\times$ scale. The previous spatial down-sampling community stands on the static weighting templates and is not relevant with the image context. By contrast, inspired by the signal sampling theorem, we parameterize the static weighting down-sampling operator as a learnable and context-adaptive operator in a unified Fourier function.

used types including the ideal low-pass filter that truncates high frequencies in the Fourier domain and the Gaussian low-pass filter that gradually attenuates frequency components near the boundary. However, the ideal low-pass filter may introduce ring artifacts due to spectrum leakage, while the Gaussian low-pass filter may result in a significant loss of edge information that is crucial for visual recognition tasks.

The prevailing approaches in the down-sampling family rely on a static weighting strategy, which may lead to unintended biases (See Section 4.4 for details.). As described in Figure 1, strided convolution and strided pooling variants rely on the static template $w_i = [0.25, 0.25, 0.25, 0.25]$ over the corresponding cornered positions while the ideal low-pass one exploits the $w_i = [0.25, 0, 0, 0]$ weighting template. (See Appendix B for proofs.) All of them are shared over all the coordinated positions and uninvolved with the feature context. It is widely acknowledged that the static sampling approach, which lacks contextual relevance, is sub-optimal for visual tasks. Therefore, both bridging different down-sampling approaches and achieving an optimal approach are desirable, as shown in Figure 1 where we focus on unifying the down-sampling modeling rules in a learnable and context-adaptive parameterized function in Fourier domain.

In this study, we delve into the working mechanism of the spatial down-sampling family and analyze the biased effects resulting from the static weighting strategy used in existing down-sampling approaches. To solve the bias problem, we propose a novel down-sampling paradigm called Fouri-Down, which operates in the Fourier domain and adapts the feature sampling based on the image context. Inspired by the signal sampling theorem, we parameterize the non-parameter static weighting down-sampling operator as a learnable and context-adaptive operator in a unified Fourier function. Furthermore, drawing from this insight, we organize the corresponding frequency positions of the 2D plane, ensuring that they are physically closed in a single channel dimension. We then perform point-wise channel shuffling based on an indicator that determines whether a channel's signal frequency bin is prone to aliasing, thereby maintaining the consistency of the weighting parameter learning. Fouri-Down, as a generic operator, comprises four key components: 2D discrete Fourier transform, context shuffling rules, Fourier weighting-adaptively superposing rules, and 2D inverse Fourier transform. These components can be readily integrated into existing image restoration networks, allowing for a plug-and-play approach. To verify its efficacy, we conduct extensive experiments across multiple computer vision tasks, including image de-blurring and low-light image enhancement. The results demonstrate that FouriDown consistently outperforms the baselines, showcasing its capability of performance improvement.

In conclusion, this work propose a novel and unified framework for the research of down-sampling, which have the following contributions.

**1)** We provide the first exploration of the aliasing problem in deep neural networks, analyzing it from a spectrum perspective.

**2)** To achieve dynamic frequency aliasing, we introduce a unified approach to down-sampling strategies within the Fourier function. Additionally, we propose a learnable and context-adaptive down-sampling operator based on the Nyquist signal sampling theorem.

**3)** Our proposed down-sampling approach serves as a plug-and-play operator, consistently enhancing the performance of image restoration tasks, such as low-light enhancement and image deblurring.

## 2 Related Work

### 2.1 Traditional Down-Sampling

Downsampling is an important and common operator in computer vision [14, 15, 16, 17, 18, 19, 20, 21, 22, 23, 24, 25], which benefits from enlarging the receptive field and reducing computational costs. So many models incorporate downsampling to allow the primary reconstruction components conducting at a lower resolution. Moreover, with the emergence of increasingly compute-intensive large models, downsampling becomes especially crucial, particularly for high-resolution input images.

Previous downsampling methods often utilized local spatial neighborhood computations (e.g., Bilinear, Bicubic and MaxPooling), which show decent performances across various tasks. However, these computations are relatively fixed, making it challenging to maintain consistent performance across different tasks. To address this, some methods made specific designs to make the downsampling more efficient in specific tasks. For instance, some works [12, 11, 10, 7]introduce the Gaussian blur kernel before the downsampling convolution to combat aliasing for better shift-invariance in classification tasks. Grabinski et al. [26, 27] equip the ideal low-pass filter or the hamming filter into downsampling to enhance model robustness and avoid overfitting.

### 2.2 Dynamic Down-Sampling

Due to the development of data-driven deep learning, in addition to traditional down-sampling, some other works [28, 29, 30, 31, 32, 33] introduce dynamic downsampling to adaptively adjust for different tasks, thereby achieving better generalizability. For instance, Pixel-Shuffle [28] enables dynamic spatial neighborhood computation through the interaction between feature channels and spaces, restoring finer details more effectively. Kim et al. [29] proposes a task-aware image downsampling to support upsampling for more efficient restoration.

In addition to dynamic neighborhood computation, dynamic strides have also gained widespread attention in recent years. For instance, Riad et al. [30] posits that the commonly adopted integer stride of 2 for downsampling might not be optimal. Consequently, they introduce learnable strides to explore a better trade-off between computation costs and performances. However, the stride is still spatially uniformly distributed, which might not be the best fit for images with uneven texture density distributions. To address this issue, dynamic non-uniform sampling garners significant attention [31, 32, 33]. For example, Thavamani et al. [31] proposed a saliency-guided non-uniform sampling method aimed at reducing computation while retaining task-relevant image information.

In conclusion, most of recent researches focus on dynamic neighborhood computation or dynamic stride for down-sampling, where the paradigm can be represented as $Down(s)$, where $s$ denotes the stride. However, in this work, we observe that the methods based on this downsampling paradigm employ static frequency aliasing, which may potentially hinder further development towards effective downsampling. However, learning dynamic frequency aliasing upon the existing paradigm poses challenges. To address this issue, we revisit downsampling from a spectral perspective and propose a novel paradigm for it, denoted as $FouriDown(s,w)$. This paradigm, while retaining the stride parameter, introduces a new parameter, $w$, which represents the weight of frequency aliasing during downsampling and is related to strides. Further, based on this framework, we present an elegant and effective approach to achieve downsampling with dynamic frequency aliasing, demonstrating notable performance improvements across multiple tasks and network architectures.

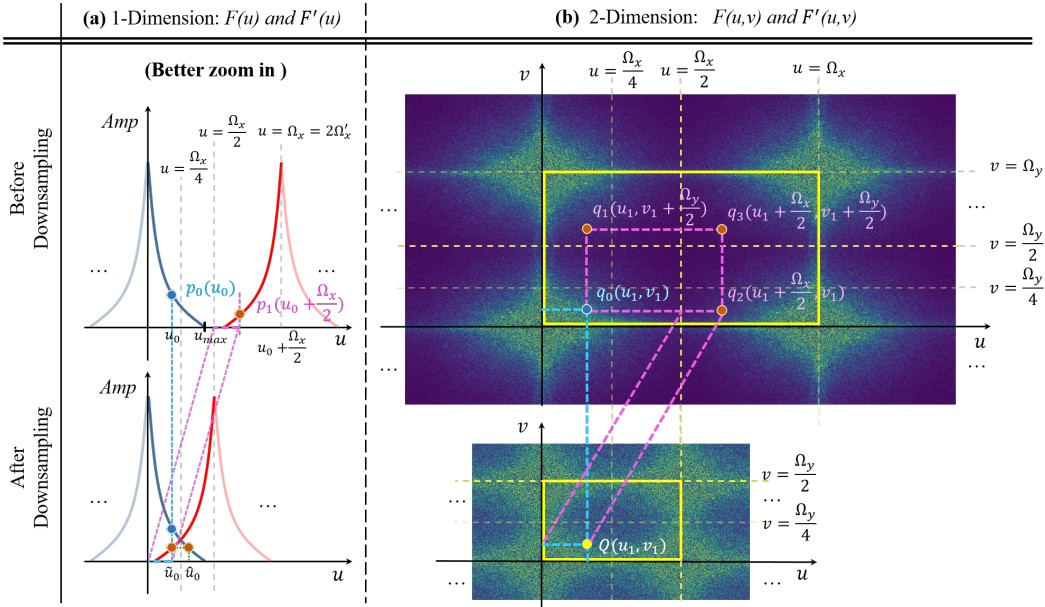

Figure 2: The visualization of the shuffling and superposing theory in the 1-D and 2-D signals.

## 3 Method

**Definitions.** $f(x, y) \in \mathbb{R}^{\text{H} \times \text{W} \times \text{C}}$ is the 2-D discrete spatial signal and the sampling rates in $x$ and $y$ axis are $\Omega_x$ and $\Omega_y$, respectively. $F(u, v) \in \mathbb{R}^{\text{H} \times \text{W} \times \text{C}}$ is the Fourier transform of $f(x, y)$, where the maximum frequencies in $u$ and $v$ axis are respectively denoted as $u_{max}$ and $v_{max}$. Moreover, $f'(x, y) \in \mathbb{R}^{\frac{\text{H}}{2} \times \frac{\text{W}}{2} \times \text{C}}$ is 2-strided down-sampled f(x,y) and its Fourier transform $F'(u, v)$.

**Theorem-1. Shuffling and Superposing.** *The spatial down-sampling typically results in a shrinkage of the tolerance for the maximum frequency of the signal. Specifically, high frequencies will fold back into new low frequencies and superpose onto the original low frequencies. To illustrate with 1-dimensional signal, the high and low frequency superposition in the down-sampling can be formulated as*

$$F'(u) = \mathbb{S}(F(u), F(u + \frac{\Omega_x}{2})) \quad when \quad u \in (0, \frac{\Omega_x}{2}), \tag{1}$$

*where $\mathbb{S}$ is a superposing operator. Note that the high frequency is $F(u + \frac{\Omega_x}{2})$ considering positive directions, while the low frequency is $F(u)$ considering positive directions instead.*

**Theorem-2. Static Averaging Superposing.** *For an image, the spatial down-sampling operator with 2 strides can be equivalent to dividing the Fourier spectrum into $2 \times 2$ equal parts and superposing them averagely by $\frac{1}{4}$ factor*

$$F(u, v) = \left[ \begin{array}{c|c} F_{(0,0)}(u, v) & F_{(0,1)}(u, v) \\ \hline F_{(1,0)}(u, v) & F_{(1,1)}(u, v) \end{array} \right], \tag{2}$$

*where $F_{(i,j)}(u, v)$ is a sub-matrix of $F(u, v)$ by equally dividing $F(u, v)$ into $2 \times 2$ partitions and $i, j \in \{0, 1\}$. Given that $\text{Down}_2$ is 2-strided down-sampling operator and $\text{IDFT}$ is inverse discrete Fourier transform, we have*

$$\text{Down}_2(f(x, y)) = \text{IDFT} \left( \frac{1}{4} \sum_{i=0}^{1} \sum_{j=0}^{1} F_{(i,j)}(u, v) \right). \tag{3}$$

The proof of the above theorem can be found in the Appendix, and examples of 1-D and 2-D signals can also be referred in Figure 2(a) and (b).

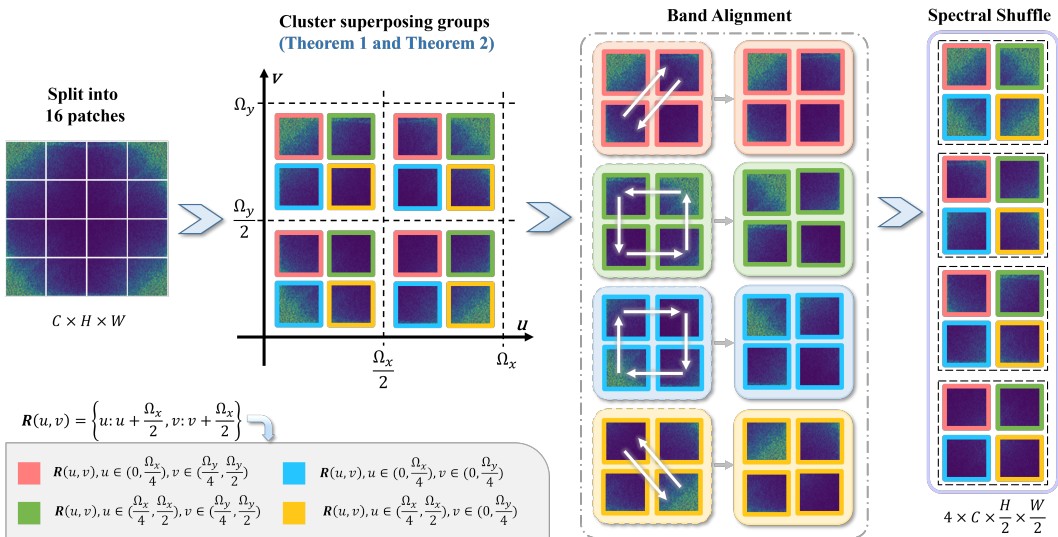

Figure 3: Overview of the proposed spectral shuffle.

---

**Algorithm 1** Pseudo-code of FouriDown.

---

**Data:** Input: $x \in \mathbb{R}^{N \times C \times H \times W}$. Set of real numbers: $\mathbb{R}$. Set of complex numbers: $\mathbb{C}$.
**Result:** $y \in \mathbb{R}^{N \times C \times \frac{H}{2} \times \frac{W}{2}}$

$x_{fft} \leftarrow FFT(x) \in \mathbb{C}^{N \times C \times H \times W}$ // Fast Fourier transform
$x_{shuffle} \leftarrow Shuffle(x_{fft}) \in \mathbb{C}^{N \times 4 \times C \times \frac{H}{2} \times \frac{W}{2}}$
$x_{weight} \leftarrow Softmax(Convs_{1x1}(x_{shuffle}, groups = C)) \in \mathbb{C}^{N \times 4 \times C \times \frac{H}{2} \times \frac{W}{2}}$
$x_{superposed} \leftarrow Sum(x_{weight} * x_{shuffle}, dim = 1) \in \mathbb{C}^{N \times C \times \frac{H}{2} \times \frac{W}{2}}$
$y \leftarrow iFFT(x_{superposed})$ // Inverse Fourier transform

---

### 3.1 Architecture Design

In this work, we argue that the the static superposing strategy like the stride-based down-sampling in Theorem-2 might lead to biased effects. Motivated by adaptively learning ability of CNNs, we aim to parameterize the non-parameter static superposing step as a learnable and context-adaptive operator in the Fourier domain.

**Definition-2 (Shuffle-Invariance)** Given an operator $z(.)$ that is shuffle-invariant and $o_1, o_2, o_3, o_4$ as different components, the shuffle-invariant is defined as $z(o_1, o_2, o_3, o_4) = z(shuffle(o_1, o_2, o_3, o_4))$, where $shuffle(.)$ is shuffling the order of input components arbitrarily.

Note that the average operator in Theorem-2 is shuffle-invariant. For example, $Aver(a, b, c, d) = Aver(b, a, c, d)$). However, different from averaging, the convolution operator, which is sensitive to the input order, does not have this property.

To alleviate this problem, we design a spectral-shuffle strategy that first performs shuffling according to Theorem-1 and then aligns across different frequency bands, as shown in Figure 3. Specifically, we initially spilt the original spectrum $F(u, v)$ into 16 patches equally. Then, according to Theorem-1, we classify these patches into 4 group, where each group is pixel-wise matched frequency bin for superposing. However, the energy distribution in each group is different. Considering the shuffle-variance of convolution operators, we reorder the intra-group sequence for inter-group alignment. The alignment is motivated by wavelet theory, where intra-group frequencies are reordered according to low-frequency and high-frequency in horizontal direction, high-frequency in vertical direction, and high-frequency in diagonal direction. Then, the aligned groups are sorted orderly on channels for pixel-wise matching in the channel dimension. Finally, we perform adaptively weighted superposition on channels by learned weights for the down-sampling results. The main implementation is depicted in Algorithm 1. Code will be public.

# 4 Experiments and Discussion

To validate the efficacy of our proposed FouriDown, we execute comprehensive experiments across several computer vision tasks and conduct exhaustive ablation studies to evaluate its resilience.

## 4.1 Experimental Settings

**Image enhancement.** For image enhancement, we assess our FouriDown model using the LOL [34] and Huawei [35] benchmarks. The LOL dataset contains 500 image pairs (485 for training, 15 for testing), and the Huawei dataset contains 2480 paired images (2200 for training, 280 for testing). We compare our results with two established baselines, SID [36] and DRBN [37].

**Image deblurring.** For image deblurring, we utilize DeepDeblur [38] and MPRNet [39] on the DVD dataset [40], which includes 2103 training and 1111 test pairs. We further validate our model's generalizability using the HIDE dataset [41].

**Image denoising.** In the context of image de-noising, our training involves the SIDD dataset [42]. Subsequent performance assessments are carried out on the remaining validation samples from the SIDD dataset and on the DND benchmark dataset [43]. For comparative analysis, we choose baselines such as MIRNet [44], and Restormer [45].

**Image dehazing.** For image dehazing, we employ RESIDE dataset for evaluations. We also use two different network designs MSBDN [46] and GridNet [47] with our proposed operator for validation.

## 4.2 Implementation Details

Regarding the above competitive baselines, we perform the comparison over the following configurations by replacing the down-sampling operator, such as strided convolution and strided pooling), with the proposed FouriDown operator. Additionally, we also perform comparisons with previous anti-aliasing down-sampling methods, including Gaussian filter [7] and "ideal" Low-Pass Filter (LPF) [13], which conduct the static modulation on the spectrum.

1) **Original**: The original down-sampling in the baseline;
2) **Gaussian**: Following [7], equipping the Gaussian filter in all channels before the original down-sampling for anti-aliasing;
3) **LPF**: Following [13], equipping the "ideal" Low-Pass Filter in all channels before the original down-sampling operator for anti-aliasing;
4) **Ours**: Replacing our proposed FouriDown with the original down-sampling operator;

Table 1: Image enhancement comparison.

| Method | Config | LOL | | Huawei | |
|---|---|---|---|---|---|
| | | PSNR | SSIM | PSNR | SSIM |
| DRBN | Original | 19.92 | 0.7712 | 20.21 | 0.6742 |
| | Gaussian | 20.21 | 0.8146 | 20.66 | 0.6955 |
| | LPF | 18.91 | 0.7441 | 20.34 | 0.6812 |
| | Ours | 21.64 | 0.8513 | 21.46 | 0.7213 |
| SID | Original | 21.46 | 0.8584 | 20.38 | 0.6931 |
| | Gaussian | 21.78 | 0.8612 | 20.52 | 0.6926 |
| | LPF | 20.74 | 0.8124 | 20.54 | 0.6841 |
| | Ours | **23.28** | **0.8708** | **20.90** | **0.7002** |

Table 2: Image deblurring comparison.

| Method | Config | DVD | | HIDE | |
|---|---|---|---|---|---|
| | | PSNR | SSIM | PSNR | SSIM |
| DeepDeblur | Original | 29.32 | 0.8817 | 29.60 | 0.8849 |
| | Gaussian | 29.36 | 0.8823 | 29.62 | 0.8892 |
| | LPF | 29.19 | 0.8751 | 29.51 | 0.8851 |
| | Ours | **29.44** | **0.8856** | **29.70** | **0.8904** |
| MPRNet | Original | 30.12 | 0.8958 | 30.04 | 0.8945 |
| | Gaussian | 30.23 | 0.8922 | 30.06 | 0.8966 |
| | LPF | 30.00 | 0.8918 | 29.95 | 0.8937 |
| | Ours | **30.31** | **0.8996** | **30.25** | **0.9102** |

Table 3: Image de-noising comparison.

| Method | Config | SIDD | | DND | |
|---|---|---|---|---|---|
| | | PSNR | SSIM | PSNR | SSIM |
| Restormer | Original | 39.41 | 0.9171 | 39.67 | 0.9173 |
| | Gaussian | 39.43 | 0.9169 | 39.69 | 0.9177 |
| | LPF | 39.35 | 0.9162 | 39.64 | 0.9167 |
| | Ours | **39.47** | **0.9174** | **39.73** | **0.9180** |
| MIRNet | Original | 39.52 | 0.9182 | 39.41 | 0.9146 |
| | Gaussian | 39.55 | 0.9184 | 39.45 | 0.9148 |
| | LPF | 39.49 | 0.9179 | 39.35 | 0.9141 |
| | Ours | **39.64** | **0.9186** | **39.56** | **0.9251** |

Table 4: Image dehazing comparison.

| Method | Config | Indoor | | Outdoor | |
|---|---|---|---|---|---|
| | | PSNR | SSIM | PSNR | SSIM |
| MSBDN | Original | 29.77 | 0.9591 | 28.88 | 0.9581 |
| | Gaussian | 30.09 | 0.9607 | 28.91 | 0.9583 |
| | LPF | 29.92 | 0.9598 | 29.03 | 0.9591 |
| | Ours | **30.19** | **0.9612** | **29.21** | **0.9604** |
| GridNet | Original | 30.16 | 0.9616 | 29.54 | 0.9605 |
| | Gaussian | 30.21 | 0.9617 | 29.62 | 0.9622 |
| | LPF | 30.18 | 0.9611 | 29.58 | 0.9615 |
| | Ours | **30.42** | **0.9654** | **29.71** | **0.9641** |

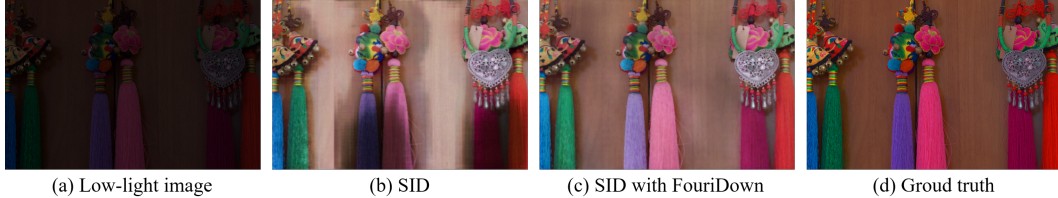

| (a) Low-light image | (b) SID | (c) SID with FouriDown | (d) Groud truth |

Figure 4: Visual comparison of SID on the LOL dataset. FouriDown enhances the global color perception ability of the original model, thereby improving the model's performance without adding extra parameters or computational overhead.

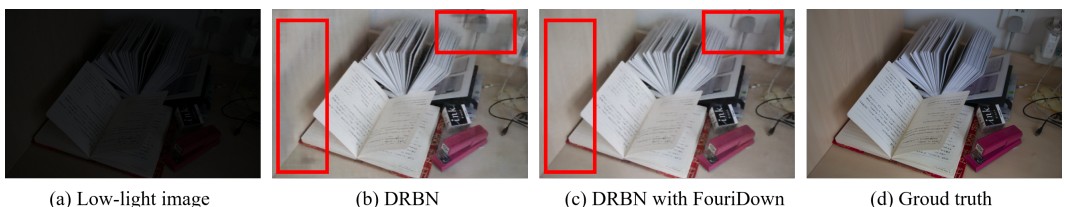

| (a) Low-light image | (b) DRBN | (c) DRBN with FouriDown | (d) Groud truth |

Figure 5: Visual comparison of DRBN on the LOL dataset. The more flexible frequency interaction mechanism in FouriDown reduces artifacts compared to the original methods.

### 4.3 Comparison and Analysis

**Quantitative Comparison.** To demonstrate the effectiveness of our proposed FouriDown, we conduct extensive experiments as shown in Tables 1-4. The best results are in bold. Above and below the baseline are highlighted in red and blue, respectively. From these tables, although previous anti-aliasing methods may be useful for some image restoration tasks, their static weights limit their universality in other tasks. For instance, while the LPF approach performs well in dehazing, it fails to deliver effective in deblurring and low-light enhancement. In contrast, our method is proved to be effective across the majority of image restoration tasks. Specifically, we achieved an improvement of 1.82dB in low-light enhancement and 0.42dB in dehazing on LOL and Reside dataset respectively.

Further, we compare the computing costs with other methods shown in Table 5. We include results from traditional down-sampling techniques like bicubic, bilinear, pixel-unshuffle, 2x2 learnable CNN (with stride=2), max-pooling, average-pooling, LPF, Gaussian and Ours. Noting that the "Original" down-sampling of the method is pointed by asterisk ('*'). This will allow a clearer contrast and showcase the advantages of our method not only against anti-aliasing approaches but also against these conventional down-sampling methods.

**Qualitative Comparison.** Due to space constraints, we only present a qualitative comparison on the low-light enhancement task. As illustrated in Figure 4 and Figure 5, our FouriDown reduces original artifacts presented in the SID due to the more flexible frequency interactions. Then, we compare the visualizations of the feature maps and their corresponding spectra between FouriDown and other down-sampling methods (see Figure 6 and Figure 7). It can be observed that the model equipped with FouriDown generates much stronger responses to degradation-aware regions, i.e. global low-illumination in the low-light enhancement task. In contrast, the model with other down-sampling method responds weakly to these regions. The results demonstrates the effectiveness of FouriDown in capturing degradation-aware information by adaptive frequency superposition in down-sampling. For the Gaussian method, its response to degradation is relatively large (second only to FouriDown), thus achieving performance that is also second only to FouriDown. Similarly, as the LFP method has the poorest performance, its feature response of the low-light areas is also the lowest. The performance of other methods is roughly similar, so their feature responses are also quite similar, indicating a similar capability to capture image degradation areas. Additionally, from the spectral comparison in Figure 7, it can be observed that the Gaussian method loses a lot of high-frequency information compared to FouriDown. This leads to challenges in recovering textures in dark areas. Hence, although the Gaussian method exhibits good responses, FouriDown achieves better performances compared to it. More qualitative comparisons can be found in the following supplementary material.

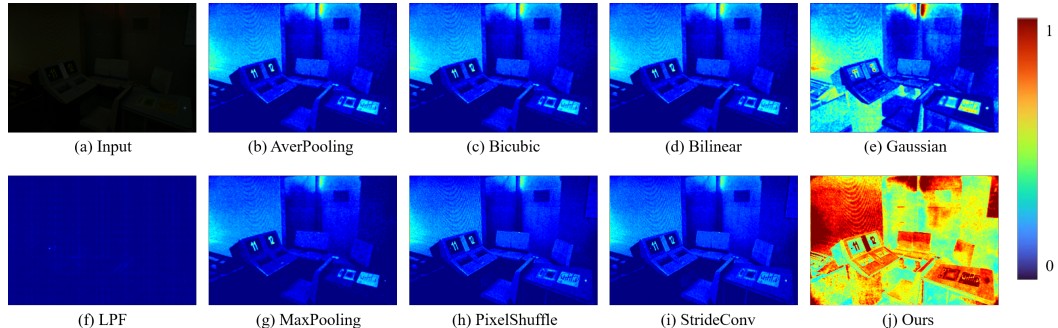

Figure 6: Feature comparison between our FouriDown and other down-sampling methods in low-light enhancement task. Due to the unique global modeling mechanism in the frequency domain, the features extracted by our method significantly achieve a larger response than others.

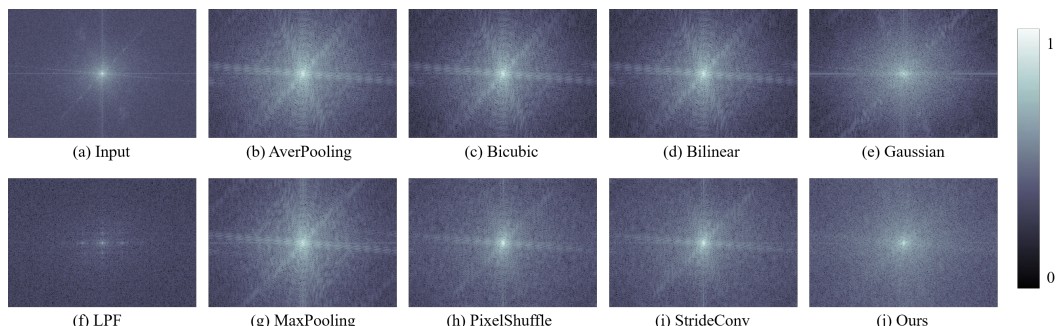

Figure 7: Spectrum comparison of the feature maps in Figure 6. The spectrum following FouriDown obtains the outstanding smooth response in both high and low frequencies.

### 4.4 Discussions

**Bias Effects by Static Superposing.** As shown in Figure 8, we compare different down-sampling methods with static superposing manner, and find they have various bias effects.

**Frequency Aliasing Visualization.** To delve deeper into the high-low frequency interactions in down-sampling, we examine the spectrum of images down-sampled by factors of 4x, 2x, and 1x. Following Theorem 1, some regions of spectrums are overlaid on the same frequency band, with smaller scales overlaying larger ones, as shown in Figure 9. This alignment of same bandwidth reveals a rectangular contour at the intersections, where high-frequencies not obeying the Nyquist theory fold into low frequencies during down-sampling, as pointed by the yellow arrow. This suggests that it is significant for down-sampling to modulate frequencies, otherwise it might degrade the original signal undesirably.

**Other Discussions.** Because of space constraints, for more discussions, including extensions to Theorem-2 and revisiting of previous anti-aliasing methods in the proposed FouriDown framework, could be referred to the supplementary material.

## 5 Limitations

In this work, we explore spatial down-sampling from a frequency-domain perspective and optimize the static weighting of previous down-sampling with a stride of 2 in the frequency domain. Our modeling of down-sampling is based on using uniformly distributed impulse sequences as the sampling function, hence exploring the characteristics of the sampling function in the frequency domain. However, for non-uniform down-sampling, where the sampling rate varies according to the content, our method might become limited. We hope to overcome this limitation in the future work by exploring the frequency domain response of non-uniform sampling functions.

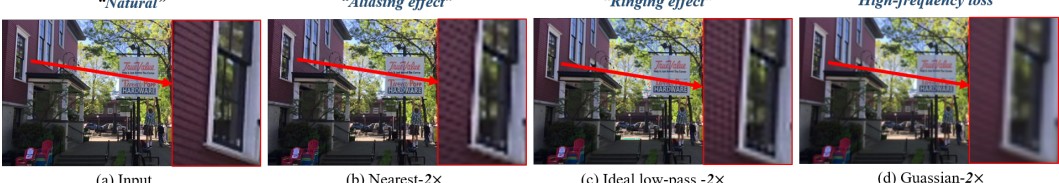



"Natural"   "Aliasing effect"   "Ringing effect"   "High-frequency loss"

(a) Input     (b) Nearest-*2×*     (c) Ideal low-pass -*2×*     (d) Guassian-*2×*



Figure 8: The comparison of results by different down-sampling manner in $2x$ scale. In (b), which is the nearest down-sampling of (a), some unnatural stripes in the edge of windows and walls, called aliasing effects. To relieve the occurrence of aliasing, (c) and (d) employ the "ideal low-pass filter" and the Gaussian filter before down-sampling respectively. However, these manners usually lead to ringring effects or heavy high-frequency loss inevitably.

| Config | LOL | | FLOPs(G) | Para(M) |
| --- | --- | --- | --- | --- |
| | PSNR | SSIM | | |
| Bicubic | 21.35 | 0.8497 | 13.764 | 7.84 |
| Bilinear | 21.26 | 0.8464 | 13.764 | 7.84 |
| Pixle-shuffle | 21.41 | 0.8552 | 13.954 | 8.11 |
| Stride Conv | 21.36 | 0.8534 | 13.954 | 8.11 |
| Max pooling * | 21.46 | 0.8584 | 13.753 | 7.84 |
| Average pooling | 21.34 | 0.8481 | 13.754 | 7.84 |
| Gaussian | 21.79 | 0.8612 | 16.102 | 8.54 |
| LPF | 20.74 | 0.8124 | 16.102 | 8.54 |
| Ours | 23.28 | 0.8708 | 13.827 | 7.87 |

Table 5: The effectiveness and efficiency comparison between our FouriDown and other down-sampling methods in low-light enhancement task.

Figure 9: The spectrums of different-scale images placed under the same frequency coordinate system.

## 6 Conclusion

In our study, we revisit the spatial down-sampling techniques and anti-aliasing strategies from a Fourier domain perspective, recognizing their reliance on static high and low frequency superposing. As a result, we propose a novel approach (FouriDown) to learn a learnable frequency-context interplay among high and low frequencies during down-sampling. Moreover, this work is the first exploration of dynamic frequency interaction in down-sampling. The FouriDown is designed based on the signal sampling theory, so it is convenient to replace most of current down-sampling and anti-aliasing techniques. Extensive experiments demonstrate the performance improvements across a variety of vision tasks.

Ultimately, we believe that down-sampling is a crucial research direction in the future. It allows for network design at a lower resolution, significantly reducing the computational overhead and enabling light-weighting models.

## Broader Impact

This work further exploits the potential of down-sampling in the Fourier domain and offers a new perspective that frequency band shuffling and superposing for future research of down-sampling. Down-sampling techniques can potentially make the future model development more efficient and effective, beneficial for various machine learning and AI applications. Nonetheless, the efficacy of our method could be a source of concern if not properly utilized, especially in terms of the safety of real-world applications. To alleviate such concerns, we plan to rigorously investigate the robustness and effectiveness of our approach across a more diverse spectrum of real-world scenarios.

## Acknowledgments

This work was supported by the JKW Research Funds under Grant 20-163-14-LZ-001-004-01, and the Anhui Provincial Natural Science Foundation under Grant 2108085UD12. We acknowledge the support of GPU cluster built by MCC Lab of Information Science and Technology Institution, USTC.

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

## Appendix A: Proofs of Theorem-1 and Theorem-2

**Proof of Theorem-1: Shuffling and Superposing**

To model the relationship between $f'(x)$ and $f(x)$, we stand on their derived continuous signal $g(x)$ by a specific sampling function. Note that the sampling functions $s_{\Delta T}(x)$ is defined as the sum of infinitely impulses units separated by $\Delta T$ intervals:

$$s(x, \Delta T) = \sum_{n=-\infty}^{\infty} \delta(x - n\Delta T). \tag{4}$$

Based on the Eq. (4), $f(x)$ and its down-sampling $f'(x)$ can be represented as

$$f(x) = g(x)s(x, \Delta T), \quad f'(x) = g(x)s(x, 2\Delta T). \tag{5}$$

According to the Fourier transform and convolution theorem, the aforementioned spatial sampling can be expressed in the Fourier domain as:

$$F(u) = G(u) \star S(u, \Delta T) = \int_{-\infty}^{\infty} G(\tau)S(u - \tau, \Delta T)d\tau$$
$$= \frac{1}{\Delta T} \sum_{n}^{\infty} \int_{-\infty}^{\infty} G(\tau)\delta\left(u - \tau - \frac{n}{\Delta T}\right) d\tau = \frac{1}{\Delta T} \sum_{n}^{\infty} G\left(u - \frac{n}{\Delta T}\right), \tag{6}$$

where $G(u)$ and $S(u, \Delta T)$ are the Fourier transform of $g(x)$ and $s(x, \Delta T)$. From Eq. (6), it can be observed that the spatial sampling introduces the periodicity to the spectrum and the period is $\frac{1}{\Delta T}$.

Note that the sampling rates of $f(x)$ and $f'(x)$ are $\Omega_x$ and $\Omega'_x$, the relationship between them can be written as

$$\Omega_x = \frac{1}{\Delta T}, \quad \Omega'_x = \frac{1}{2\Delta T} = \frac{1}{2}\Omega_x. \tag{7}$$

Before down-sampling, to focus on the following down-sampling operation, we assume that $f(x)$ adheres to the Nyquist sampling theorem, which implies that $u_{max} > \frac{1}{\Omega_x}$.

After down-sampling, according to Nyquist sampling theorem, a whole sub-frequency band is limited in $(0, \frac{\Omega_x}{2})$. Moreover, the resulted band is the superposition of two original bands, which denoted as

$$F'(u) = \mathbb{S}(F(u), F(\hat{u})), \tag{8}$$

where $\hat{u}$, $u$ are the high frequency above the sampling rate and the low frequency below the sampling rate, respectively, and $\mathbb{S}$ is superposition operator.

**(1)** In the positive sub-band, where $u \in (0, \frac{\Omega_x}{4})$, $\hat{u}$ and $\tilde{u}$ should satisfy

$$u \in (0, \frac{\Omega_x}{4}) \quad and \quad \hat{u} \in (\frac{\Omega_x}{4}, u_{max}). \tag{9}$$

According to the aliasing theorem, the high frequency $\hat{u}$ is folded back to the low frequency:

$$\tilde{u} = \left| \hat{u} - \frac{(k + 1)\Omega'_x}{2} \right|, \quad \frac{k\Omega'_x}{2} \leq \hat{u} \leq \frac{(k + 2)\Omega'_x}{2} \tag{10}$$

where $k = 1, 3, 5 \ldots$ and $\tilde{u}$ is folded results by $\hat{u}$.

According to Eq. (9) and Eq. (10), we have

$$\tilde{u} = \frac{\Omega_x}{2} - \hat{u}, \quad and \quad \tilde{u} \in (\frac{\Omega_x}{2} - u_{max}, \frac{\Omega_x}{4}). \tag{11}$$

Then, according to Eq. (8) and Eq. (11), we attain

$$F'(u) = \begin{cases} F(u) & \text{if } u \in (0, \frac{\Omega_x}{2} - u_{max}), \\ \mathbb{S}\left(F(u), F(\frac{\Omega_x}{2} - u)\right) & \text{if } u \in (\frac{\Omega_x}{2} - u_{max}, \frac{\Omega_x}{4}). \end{cases} \tag{12}$$

According to Eq. (6), $F(u)$ is symmetric with respect to $u = \frac{\Omega_x}{2}$,

$$F(\frac{\Omega_x}{2} - u) = F(u + \frac{\Omega_x}{2}). \tag{13}$$

Further, $F(u + \frac{\Omega_x}{2}) = 0$ when $u \in (0, \frac{\Omega_x}{2} - u_{max})$. Upon Eq. (13), we can uniform Eq. (12) as

$$F'(u) = \mathbb{S}(F(u), F(u + \frac{\Omega_x}{2})) \quad when \quad u \in (0, \frac{\Omega_x}{4}) \tag{14}$$

The visualization of the aforementioned proof process is depicted in Figure 2(a).

**(2)** In the negative sub-band, where $u \in (\frac{\Omega_x}{4}, \frac{\Omega_x}{2})$, different from (1), $\hat{u}$ and $\tilde{u}$ should satisfy

$$u \in (\frac{\Omega_x}{4}, \frac{\Omega_x}{2}) \quad and \quad \hat{u} \in (\frac{\Omega_x}{2} - u_{max}, \frac{\Omega_x}{4}). \tag{15}$$

Similarly, we can proof $F(u)$ in the negative sub-band as well.

$$F'(u) = \mathbb{S}(F(u), F(u + \frac{\Omega_x}{2})) \quad when \quad u \in (\frac{\Omega_x}{4}, \frac{\Omega_x}{2}) \tag{16}$$

Combined with Eq. (14) and Eq. (16), we obtain

$$F'(u) = \mathbb{S}(F(u), F(u + \frac{\Omega_x}{2})), \quad when \quad u \in (0, \frac{\Omega_x}{2}). \tag{17}$$

**Proof of Theorem-2: Static Superposing.**

According to Eq. (7) and Eq. (6), we can deduce that the amplitude of $F'$ is half that of $F$.

Therefore, we can write $F'(u)$ in the $x$ axis as,

$$F'(u) = \frac{1}{2}F(u) + \frac{1}{2}F(u + \frac{\Omega_x}{2}), \quad when \quad u \in (0, \frac{\Omega_x}{2}). \tag{18}$$

Upon the dual principle, we can prove $F'(v)$ in the whole sub-band

$$F'(u, v) = F'(F'(u, y), v) = \frac{1}{4}\left(F(u, v) + F(u + \frac{\Omega_x}{2}, v) + F(u, v + \frac{\Omega_y}{2}) + F(u + \frac{\Omega_x}{2}, v + \frac{\Omega_y}{2})\right),$$
$$\tag{19}$$

where $u \in (0, \frac{\Omega_x}{2}), v \in (0, \frac{\Omega_y}{2})$.

## Appendix B: Revisiting previous down-sampling manners in the FouriDown

In the main body, we introduce the FouriDown framework which can simultaneously adjust the stride of down-sampling and the characteristics of frequency interaction. Moreover, in the manuscript, we prove that the operator with a stride of 2 can be simplified to a fixed frequency weighting, *i.e.* averaging. However, this kind of weighting brings aliasing, hence the need for anti-aliasing methods such as ideal low-pass filters and Gaussian filters, which put focused weighting on frequencies. In this section, we revisit these anti-aliasing methods within the FouriDown framework and discover that their weighting methods remain fixed. In other words, within our framework, it's possible to realize the aforementioned anti-aliasing methods by simply changing specific parameters as shown in Figure 10 and 11.

## Appendix C: Stride Extension of Theorem-1 and Theorem-2

In the manuscript, we propose FouriDown which primarily focuses on analyzing frequency interactions in the case of stride=2. In this section, we extend the Theorem-2 to arbitrary integer strides $s$, resulting in the following theorem. The proof for these theorems follow similar logic to that detailed in the main body, and is therefore omitted for brevity.

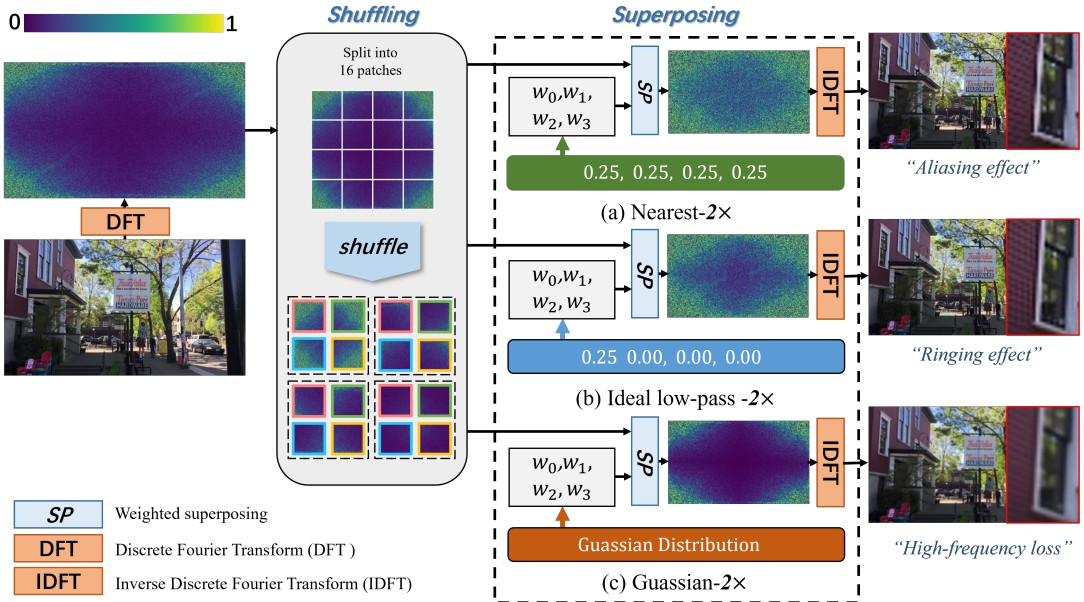

Figure 10: The achievements of different down-sampling manners in the FouriDown by simply changing specific parameters.

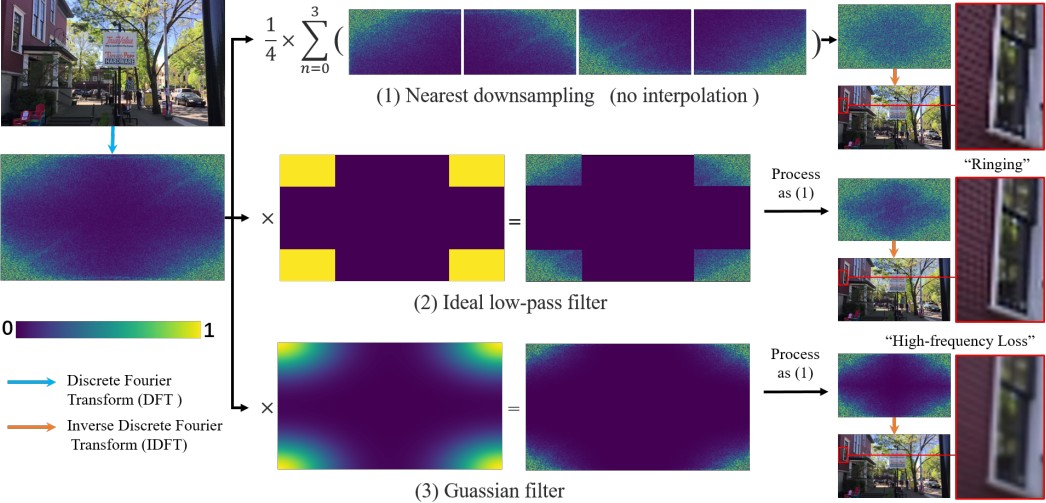

Figure 11: Revisiting different down-sampling methods from the perspective of spectrum.

**Stride Extension of Theorem-1.** *In the condition of $s$ strides, to illustrate with 1-dimensional signal, the high and low frequency superposition in the down-sampling can be formulated as*

$$F'(u) = \mathbb{S}(F(u), F(u + \frac{k_i \Omega_x}{s})) \quad when \quad u \in (0, \frac{\Omega_x}{2}), \tag{20}$$

*where $\mathbb{S}$ is a superposing operator and $k_i = 1, ..., s - 1$.*

**Stride Extension of Theorem-2.** *For an image, the spatial down-sampling operator with $s$ strides can be equivalent to dividing the Fourier spectrum into $s \times s$ equal parts and superposing them averagely by $\frac{1}{s^2}$ factor*

*Given that $\mathrm{Down_s}$ is $s$-strided down-sampling operator and $\mathrm{IDFT}$ is inverse discrete Fourier transform, we have*

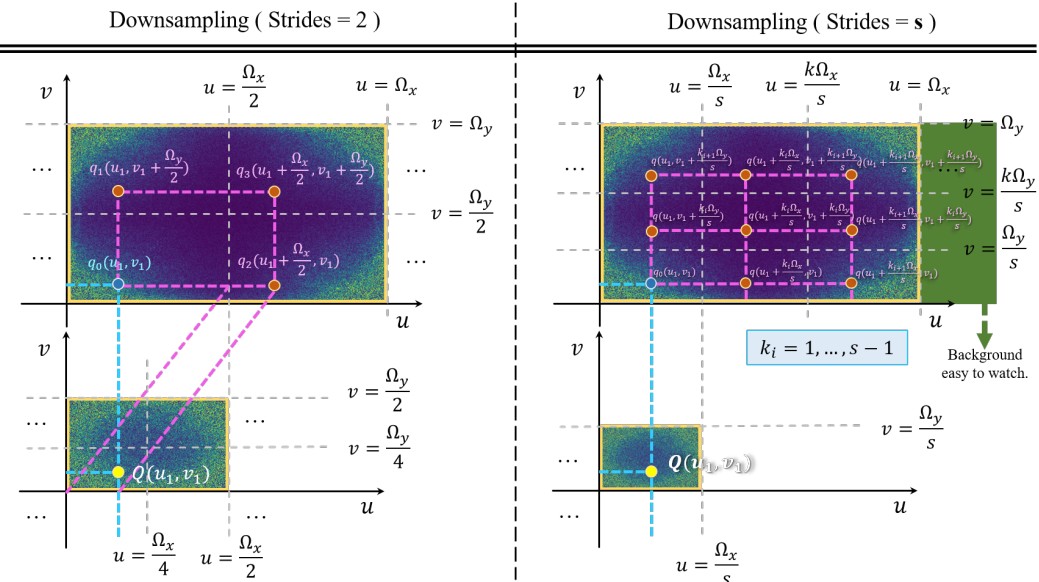

Figure 12: Illustration of theorem extension from *strides=2* to *strides=s*. Due to visualization space limitation, $s$ is taken as 3 in the figure. The background region has no meaning, just to show more clearly.

$$\text{Down}_s(f(x,y)) = \text{IDFT}\left(\frac{1}{s^2}\sum_{i=0}^{s-1}\sum_{j=0}^{s-1}F_{(i,j)}(u,v)\right). \tag{21}$$

Based on above extension, the down-sampling of 3 strides could be presented as in Figure 12.

## Appendix D: More Qualitative comparison.

Due to the limited space, we only report the visual results of the low-light enhancement task in main manuscript. We report more visual results in the supplementary materials. As shown, integrating the FouriDown with the original baseline [36, 37, 48, 49, 39, 45, 44, 46, 47] achieves more visually pleasing results as shown in Figure 13 14 15 16 17.

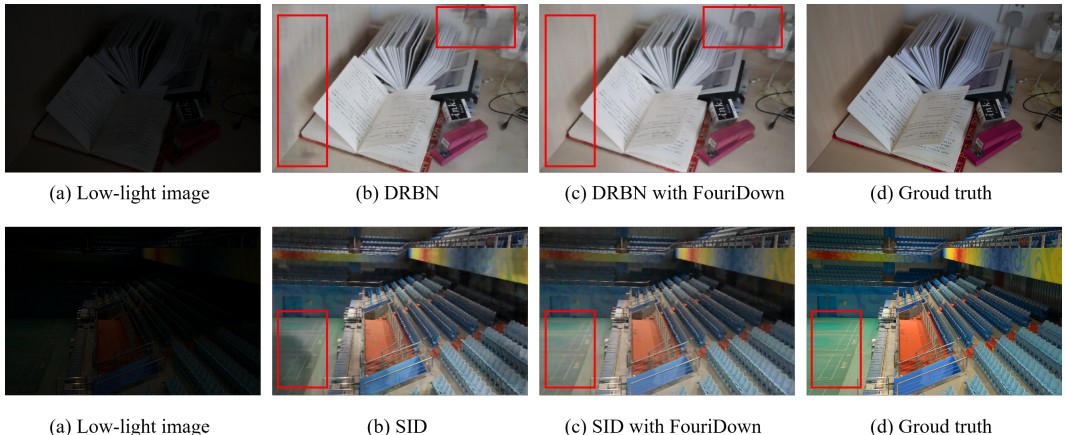

| (a) Low-light image | (b) DRBN | (c) DRBN with FouriDown | (d) Groud truth |

| (a) Low-light image | (b) SID | (c) SID with FouriDown | (d) Groud truth |

Figure 13: **Visual comparison on the low-light enhancement task.**

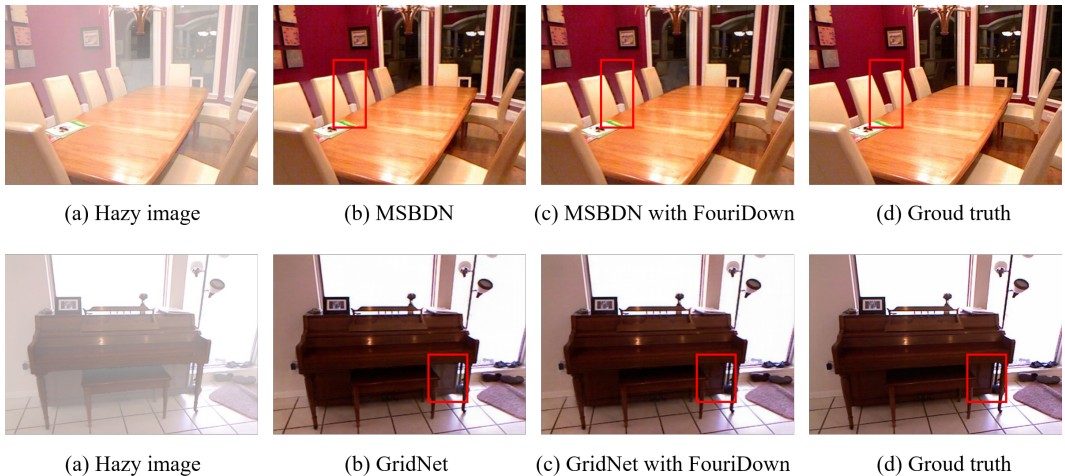

| (a) Hazy image | (b) MSBDN | (c) MSBDN with FouriDown | (d) Groud truth |

| (a) Hazy image | (b) GridNet | (c) GridNet with FouriDown | (d) Groud truth |

Figure 14: **Visual comparison on the dehazing task.**

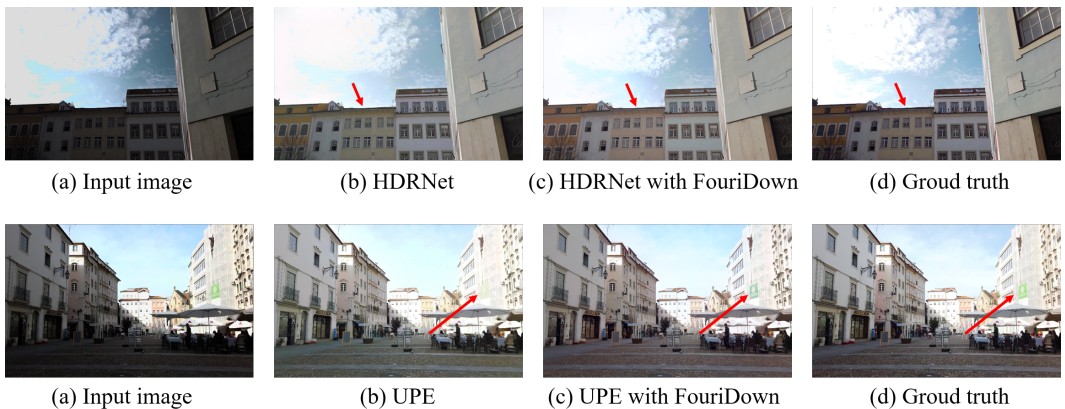

| (a) Input image | (b) HDRNet | (c) HDRNet with FouriDown | (d) Groud truth |

| (a) Input image | (b) UPE | (c) UPE with FouriDown | (d) Groud truth |

Figure 15: **Visual comparison on the UHD-enhancement task.**

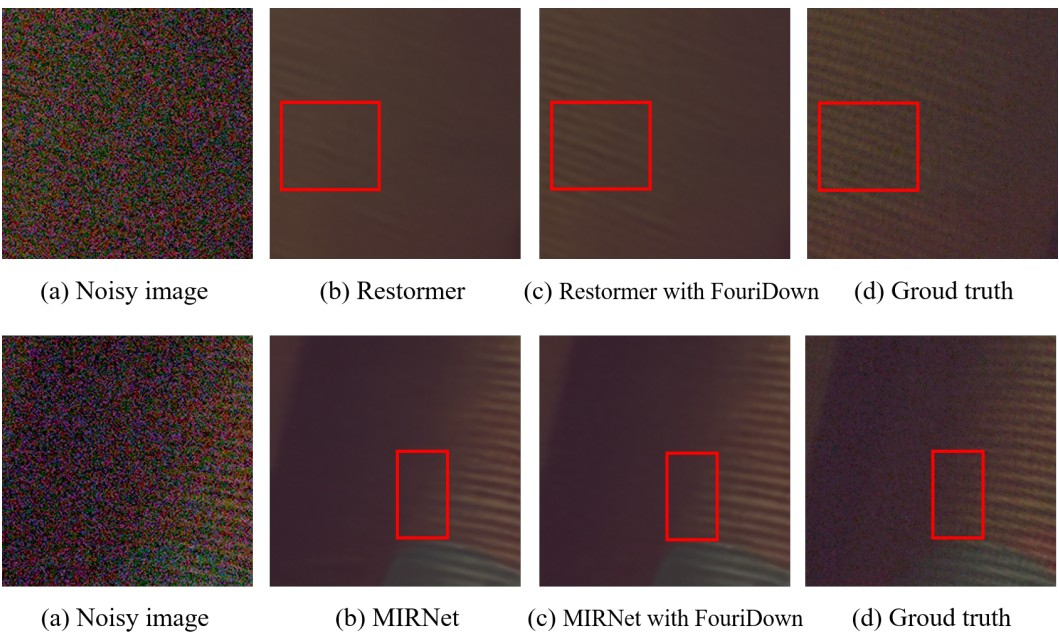

| (a) Noisy image | (b) Restormer | (c) Restormer with FouriDown | (d) Groud truth |

| (a) Noisy image | (b) MIRNet | (c) MIRNet with FouriDown | (d) Groud truth |

Figure 16: **Visual comparison on the denoising task.**

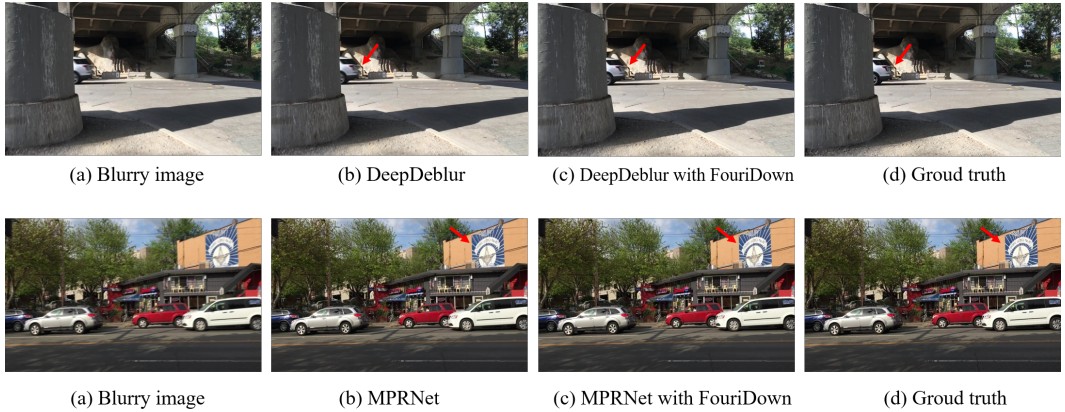

(a) Blurry image        (b) DeepDeblur        (c) DeepDeblur with FouriDown        (d) Groud truth

(a) Blurry image        (b) MPRNet        (c) MPRNet with FouriDown        (d) Groud truth

Figure 17: **Visual comparison on the deblurring task.**

