# OpenReview forum: "FouriDown: Factoring Down-Sampling into Shuffling and Superposing"
_NeurIPS.cc/2023/Conference — NeurIPS 2023 poster_

### Official Review · Reviewer_V6WG · 2023-07-05

**Soundness:** 4 excellent
**Presentation:** 4 excellent
**Contribution:** 4 excellent
**Rating:** 7
**Confidence:** 5

**Summary:**

This paper introduces a novel down-sampling operator in the Fourier domain, specifically tailored to mitigate the common problem of frequency aliasing encountered during down-sampling. The operator's efficacy has been assessed across diverse computer vision problems, consistently yielding improved performance in these tasks.

**Strengths:**

1. The authors address the long-standing issue of frequency aliasing in signal processing, with a specific focus on image down-sampling. Their meticulously crafted design offers a compelling and well-founded solution to mitigate frequency aliasing.

2. The authors substantiate their claims in the paper with comprehensive and rigorous theoretical proofs, providing a strong basis for their proposed approach.

3. The proposed FouriDown operator serves as a plug-and-play solution, seamlessly integrating into existing networks. Furthermore, it consistently enhances the performance of the underlying models across various tasks.

4. The writing in the paper is clear and the explanations are thorough, ensuring a comprehensive understanding of the presented concepts.

**Weaknesses:**

1. While the quantitative results are satisfactory, it would be beneficial to include additional efficiency analysis, such as providing the parameters of the FouriDown operator, in the form of a table showcasing relevant metrics.

2. Although the FouriDown operator is currently limited to replacing the stride 2 down-sampling operator, it would be valuable to explore whether the authors have developed other variants capable of handling more general scenarios.

3. It would be advantageous to present more visual results while condensing reducing the length of the theorem proofs.

4. Figure 5 captures attention due to its intriguing nature. It would greatly benefit the readers if the authors could provide more detailed explanations accompanying the figure to enhance understanding.

**Questions:**

Please see the weaknesses

**Limitations:**

The authors mentioned the limitations in section 5 and addressed the potential broader impact of their work.

---

> ### Author Rebuttal · Authors · 2023-08-10
>
> 1. Thank you for your sincere suggestion. We incorporating a comprehensive efficiency analysis including as following and above Table in the attached file. We will add them to further demonstrate the effectiveness and efficiency of the FouriDown operator in the revised version.
> 2. Indeed, while the current work primarily focuses on replacing stride 2 down-sampling, the underlying principle of FouriDown has the potential to be extended for more general scenarios. For example, we have explored different stride lengths in the down-sampling. As shown in Supp. Figure 6, we aim to generalize the method to accommodate varying strides, making it more adaptable to diverse applications. Moreover, beyond convolutional layers, it would be intriguing to see more potential architectures like transformers are performed for the application of specific scenarios. By delving into these areas, we hope to make the FouriDown operator even more versatile and better suited to a wider range of tasks.
> 3. Thanks for your suggestions, we will add more visualizations in the updated version.
> 4. Thank you for your suggestions. Figure 5 depicts the spectrum changes of an image as it undergoes progressive downsampling, further aiding readers in understanding the frequency aliasing process. In Figure 5, three downsampling instances are displayed, each resulting in differently sized spectral representations. These are then superimposed onto the same frequency coordinate system. According to Theorem 1, high-frequency components will fold into the lower frequencies during downsampling, so the process is illustrated by the yellow arrows in the figure. In the revised version, we will expand the figure caption and add a dedicated subsection in the manuscript.

---

> > ### Comment · Reviewer_V6WG · 2023-08-15
> > **Response to the rebuttal**
> >
> > Thank you for the response. The explanations and extra results solved my concerns.

---

### Official Review · Reviewer_rAiK · 2023-07-05

**Soundness:** 4 excellent
**Presentation:** 4 excellent
**Contribution:** 4 excellent
**Rating:** 8
**Confidence:** 5

**Summary:**

In this research, the authors re-examine the spatial down-sampling mechanism and analyze the biased impacts that arise from the static weighting strategy used in previous methods. To overcome this limitation, this work proposes a new down-sampling paradigm called FouriDown, which incorporates frequency-context adaptive superposing. This component can be seamlessly integrated into existing image restoration networks and significantly improve performance.

**Strengths:**

This work provides an in-depth exploration of the down-sampling operator, offering a novel perspective that has the potential to significantly advance future research related to operators. Furthermore, the motivation behind this work is clear and well-founded, as evidenced by the compelling depiction in Figure 2 along with the accompanying theoretical proof. Importantly, this research showcases considerable potential across diverse visual tasks.

**Weaknesses:**

1. To better emphasize the main content of the paper, it is suggested to move the proof of formulas to the supplementary material. Furthermore, the experimental section of the main text could benefit from incorporating the information from Section 1 in the supplementary material, which illustrates the superior extensibility of this framework over other down-sampling methods.
2. Although the supplementary material includes numerous excellent visual results to showcase the superiority of the proposed method, it is recommended to supplement the updated version with visualizations of the down-sampled feature maps.


**Questions:**

1. The authors present a novel down-sampling method based on Shannon's sampling theorem. As feature maps commonly comprise multiple channels, it would be interesting to investigate the interaction patterns that emerge among these channels.
2. Based on the diagram provided, the FouriDown method seems to involve information transfer onto the channel followed by channel contraction, similar to the approach used in pixel-shuffle down-sampling. It is crucial to conduct an analysis that specifically explores and demonstrates the advantages of the FouriDown method over pixel-shuffle down-sampling. This comparative analysis will provide valuable insights into the superiority of the proposed method.


**Limitations:**

The authors have adequately addressed the limitations.

---

> ### Author Rebuttal · Authors · 2023-08-10
>
> W1. Thank you for your sincere suggestion. We agree with your suggestion and will move the proof of formulas to the supplementary material. This will allow us to incorporate more relevant experiments and analysis into the main text, highlighting the superiority of our framework over other down-sampling methods.
>
> W2. Thank you for pointing this out. We perform the comparisons with visual representations of down-sampled feature maps to better illustrate the effectiveness of our method in Figure 1-3 in the attached file. In the revised version, we will incorporate them to provide a clearer understanding of how our method performs in comparison to others.
>
> Q1. In this work, the FouriDown doesn't involve interactions among channels, they operate independently, but they do interact in the spectral domain. This is analogous to the Fourier transform, where channels also operate independently when transforming feature maps. Thanks for your insightful problems and how to establish relationships between channels in the Fourier transform is an intriguing question to explore in the future.
> Moreover, while channels are processed independently during FouriDown, their interactions within convolutional networks are still related. In low-level vision tasks, channel interactions might occur between high-frequency texture information and low-frequency color information. Through our research, we find that while there's no explicit channel interaction during down-sampling, there's a manifest high-low frequency interaction occurring in the spectral domain. This could be one of the reasons why FouriDown is effective in various low-level vision tasks.
>
> Q2. Indeed, at a superficial glance, FouriDown might appear to have similarities with pixel-shuffle down-sampling in terms of channel information transfer and contraction. However, there are distinct differences:
> 1. **Domain of Operation**: FouriDown operates in the spectral domain, leveraging the properties of the Fourier transform. In contrast, pixel-shuffle primarily works in the spatial domain.
> 2. **Inherent Mechanisms**: The shuffle process in FouriDown is rooted in signal theory and is meticulously designed for convolution operations in the spectral domain. Pixel-shuffle, on the other hand, utilizes a more straightforward neighborhood sampling mechanism.
> 3. **Potential Impact**: We believe the spectrum-shuffle isn't merely a technical novelty, it has significant implications for the develop of the Fourier theory in the low-level vision task.
> In light of your feedback, we'll bolster our paper with a more in-depth comparative analysis between FouriDown and pixel-shuffle down-sampling. This will elucidate the specific advantages and the overarching superiority of our method.

---

> > ### Comment · Reviewer_rAiK · 2023-08-14
> >
> > Thanks for your responses. The authors address my concerns.

---

### Official Review · Reviewer_k1g9 · 2023-07-05

**Soundness:** 3 good
**Presentation:** 3 good
**Contribution:** 3 good
**Rating:** 6
**Confidence:** 4

**Summary:**

This paper proposes a downsampling method in Fourier domain (FouriDown) for several low-level vision tasks and an image classification task. FouriDown is supported by theoretical background proven by the authors. While this work improves multiple vision tasks when compared to previous anti-aliasing downsampling methods, it lacks to truly demonstrate its superiority over existing other downsampling approaches due to not well-organized experiments. Moreover, exploration of other architecture is insufficient. So, despite the robust and well-presented theorems the contribution of this paper is limited.

**Strengths:**

The explanation and background of this paper are theoretically robust. Why the shuffling and superposition operations of FouriDown are constructed in that manner is clear. The learnable downsampling weight, which is unlike previous anti-aliasing methods, is technically sound.

Many low-level vision tasks have been improved by FouriDown. The tables showing experimental results are easy to understand (however, some issues of experimental composition exist as mentioned in weakness). In addition to low-level tasks, the results that an image classification task is also improved makes FouriDown more robust.

**Weaknesses:**

1. Most importantly, this work focusing only on downsampling method has limited contribution for being published at this conference. If this paper has studied other architectures (e.g., CNN’s kernel or Transformer’s self-attention) and secondarily introduced the effectiveness of FouriDown, the novel downsampling method could have delivered more importance for deep learning field. This does not mean that FouriDown is not effective. Instead, more urgent architecture designs could be considered and provided in this paper. Moreover, a study regarding how to effectively and efficiently address the hierarchical downsampled multi-scale feature maps, where downsampling itself loses informative details, is suggested to be conducted.
2. While replacement of static superposition to learnable one is interesting, the composition of this learnable weight is not sophisticated enough to be considered as a novel idea. It just follows standard 1x1 convolution, activation function, and softmax. It should be carefully considered.
3. Tabs.1,2,3,4,5,6 should include some factors with respect to efficiency: inference time, the number of parameters, etc.
4. The “Original” downsampling methods in Tabs.1,2,3,4,5,6 are not presented. The authors, therefore, should present other downsampling methods that are not anti-aliasing approaches. For example, pixel-unshuffle, 2x2 learnable CNN (with stride=2), max-pooling, or other interpolation can be considered.
5. The summation operation along subsample dimension (indicated as “4”) in the Algorithm 1 can be better demonstrated by another ablation study. As alternatives to summation, there exists average operation or weighted summation for the “4”.

**Questions:**

How about ImageNet 1K classification or object detection? Did you conduct other experiments about high-level tasks? Because the performance gains are more notable in the case of CIFAR classification than low-level tasks, focusing mainly on high-level vision recognition tasks seems better.

**Limitations:**

See weakness.


I think the additional tables, figures, and description made this paper more comprehensive to be published at NeurIPS 2023. Therefore, I would like to recommend this paper to be accepted, and will change my rating from borderline reject to weak accept.

---

> ### Author Rebuttal · Authors · 2023-08-10
>
> 1. Recently, many recent studies have been centered on downsampling [reference] as stated in the related works, thus we believe the downsampling is sufficient as a crucial research topic. Our work also has ample contributions for the following reasons:
> (1)	Our study is the first to revisit the deep downsampling operator from a frequency domain perspective.
> (2)	This approach is particularly challenging due to the significant disparity between the spatial and frequency domains (details see W2).
> (3)	In this work, we innovatively pinpoint that previous downsampling methods were mostly constrained by the issue of static frequency aliasing.
> Therefore, this work comprises ample theoretical motivation and presents a challenging solution from an unprecedented perspective, which aligns well with the preferences of this conference.
>
> Meanwhile, we appreciate your sincere suggestions for hierarchical multi-scale exploration. In fact, our work essentially taps into the causes of detail information loss. Moreover, in the supplementary material, we've discussed different scale relationships in FouriDown, which might provide insights for addressing this issue. We will explore it in the future.
>
> 2. The design of the learnable weight is both carefully-considered and theory-robust.
> It seems there might be some misunderstandings regarding the novelty of our work. We don’t merely propose a simple convolution layer replacement. Importantly, before the convolution operation, a pioneering and theoretically based spectral-shuffle in Figure 3 is meticulously incorporated, ensuring the feasibility of learnable superposition.
> Further, replacing the convolution layer directly without the shuffle would result in inferior performance, as evidenced in the following table. Hence, we have rigorously considered and trialed numerous methods to realize dynamic superposition. In response to your concerns, we'd like to highlight some insights as following:
>
> |        |                    |   |       |         |
> |--------|--------------------|---|-------|---------|
> | MPRNET | FouriDown          |   | PSNR  | SSIM    |
> |        | wo shuffle         |   | 25.43 | 0.8144  |
> |        | w pixel-shuffle    |   | 23.43 | 0.7954  |
> |        | w spectral-shuffle |   | 30.31 | 0.8996  |
>
>
> (a) Designing learnable superposition is inherently challenging.
>    (i) The pronounced nonlinear discrepancy between spatial and frequency domains makes learning downsampling directly in the frequency domain via convolutional layers especially arduous. As depicted in above table, solely relying on some nonlinear activation functions is nearly inadequate to approximate the functional relationships between spatial and frequency domains. To relieve the challenging nonlinear function, we introduce spectral-shuffle, based on Theorem 1 and Theorem 2, thus offering a viable approach for realizing learnable weights via CNNs.
>    (ii) The convolution operation is typically ill-suited for spectral domains. Conventional convolution operations are primarily designed to operate in the spatial domain, capturing local features and receptive fields. However, the frequency spectrum doesn't exhibit similar biases. Most prior works [1,2] use 1x1 convolutions that share weights across the entire spectrum, essentially learning channel-wise weights. Such a design has limited the evolution of Fourier theory in deep learning. Therefore, how to extract "pseudo-local" spectral features using convolution operators remains a challenge. Fortunately, our work uncovers the 'pseudo-local' relationship within the downsampling spectrum, denoted as $[F(u), F(u + \frac{\Omega_x}{2})]$, as described in Theorem 1. Based on the derivation, we achieve a 'pseudo-local’ extraction from spectral features via convolution operators, fostering future applications and growth of Fourier theory in deep learning.
> [1] Intriguing Findings of Frequency Selection for Image Deblurring. In AAAI, 2023.
> [2] Deep Fourier-Based Exposure Correction Network with Spatial-Frequency Interaction. In ECCV, 2022.
>
>  (b) Simplicity with effectiveness epitomizes elegance.
>    (i) "Entities should not be multiplied without necessity" – Occam's razor. Resolving existing issues in a straightforward and principled manner is intrinsically significant. While more complex implementations exist, we believe they aren't particularly necessary.
>    (ii) General algorithm is usually simple. As shown in experiments section, thanks to our simple yet potent design, the FouriDown exhibits commendable versatility across various tasks (both low-level and high-level) and diverse architectures (both CNNs and Transformers). Such a design resonates with the contemporary philosophy of General Artificial Intelligence (AGI).
>
> In fact, according to Theorem 2, within our FouriDown framework, the average operation is essentially equivalent to the stride=2 operation, meaning it aligns with the baseline of many existing methods. The weighted summation is an adaptive weight prediction we introduced. The ablation study for average versus weighted summation is presented as follows.
> | | FouriDown    |   | PSNR  | SSIM    |
> |------------|--------------|---|-------|---------|
> | DeepDeblur  | Average      |   | 29.32 | 0.8817  |
> |            | Weighted sum |   | 29.44 | 0.8856  |
> | MPRNET     | Average      |   | 30.11 | 0.8956  |
> |            | Weighted sum |   | 30.31 | 0.8996  |
>
> 3.Due to time constraints, we trained the performance of YOLO v5 on the COCO subdataset and found the improvement, as shown in the table below.
> | YOLOv5    | AP   | AR    |
> |-----------|------|-------|
> | Baseline  | 23.5 | 36.8  |
> | FouriDown | 25.3 | 32.6  |
> As mentioned, FouriDown possesses good versatility. However, since this paper primarily focuses on addressing low-level tasks, we hope more experiments will be conducted in the future to further explore its potential in high-level tasks.

---

> > ### Comment · Reviewer_k1g9 · 2023-08-14
> >
> > 1. If more related works mainly focusing on down-sampling itself can be provided, the claim of the authors becomes more compelling. This discussion should include brief explanation that can reveal each related work's  main approach in regard to down-sampling method. The existing related work section can be further refined to convey the urgency and significance of addressing down-sampling challenges more explicitly.
> >
> > What is the relevant section of supplementary material that you mentioned? Does it indicate Sec.2 Stride Extension?
> >
> > 2. Thanks to your comment, I think it is justified that the adaptive convolution in superposing of FouriDown has been carefully designed. My concern of this part is resolved. Previously, I thought that introducing 1x1 conv is too simple. Now, however, I understand why complex architectures for spectral domain are not necessary.
> >
> > Moreover, regarding the average and weighted sum operation, I apologize for my misunderstanding.
> >
> > 3. It's okay.

---

> > > ### Author Response · Authors · 2023-08-17
> > >
> > > 1.Thanks for your sincere comments. We have refined the related work as following and will add them in the revision version.
> > >
> > > Downsampling is an important and common operator in computer vision, which benefits from enlarging the receptive field and reducing computational costs. So many models incorporate downsampling to allow the primary reconstruction components conducting at a lower resolution. Moreover, with the emergence of increasingly compute-intensive large models, downsampling becomes especially crucial, particularly for high-resolution input images.
> > >
> > > Previous downsampling methods often utilized local spatial neighborhood computations (e.g., bilinear, bicubic and MaxPooling), which show decent performances across various tasks. However, these computations are relatively fixed, making it challenging to maintain consistent performance across different tasks. To address this, some methods made specific designs to make the downsampling more efficient in specific tasks. For instance,  some works [1,2,3,4] introduce the Gaussian blur kernel before the downsampling convolution to combat aliasing for better shift-invariance in classification tasks. Grabinski et al. [5,6] equip the ideal low-pass filter or the hamming filter into downsampling to enhance model robustness and avoid overfitting.
> > >
> > > Moreover, some other works [7,8,9,10,11,12] introduce dynamic downsampling to adaptively adjust for different tasks, thereby achieving better generalizability. For instance, pixel-shuffle [7] enables dynamic spatial neighborhood computation through the interaction between feature channels and spaces, restoring finer details more effectively. Kim et al. [8] proposes a task-aware image downsampling to support upsampling for more efficient restoration.
> > >
> > > In addition to dynamic neighborhood computation, dynamic strides have also gained widespread attention in recent years. For instance, Riad et al. [9] posits that the commonly adopted integer stride of 2 for downsampling might not be optimal. Consequently, they introduce learnable strides to explore a better trade-off between computation costs and performances. However, the stride is still spatially uniformly distributed, which might not be the best fit for images with uneven texture density distributions. To address this issue, dynamic non-uniform sampling garners significant attention [10,11,12]. For example, Thavamani et al. [10] proposed a saliency-guided non-uniform sampling method aimed at reducing computation while retaining task-relevant image information.
> > >
> > > In conclusion, most of recent researches focus on dynamic neighborhood computation or dynamic stride for down-sampling, where the paradigm can be represented as **Down(s)**, where ‘s’ denotes the stride. However, in this work, we observe that the methods based on this downsampling paradigm employ static frequency aliasing, which may potentially hinder further development towards effective downsampling. However, learning dynamic frequency aliasing upon the existing paradigm poses challenges. To address this issue, we revisit downsampling from a spectral perspective and propose a novel paradigm for it, denoted as **FouriDown(s,w)**. This paradigm, while retaining the stride parameter, introduces a new parameter, ‘w’, which represents the weight of frequency aliasing during downsampling and is related to strides. Further, based on this framework, we present an elegant and effective approach to achieve downsampling with dynamic frequency aliasing, demonstrating notable performance improvements across multiple tasks and network architectures.
> > >
> > > [1] Blending anti-aliasing into vision transformer. In NIPS, 2022.
> > >
> > > [2] On aliased resizing and surprising subtleties in GAN evaluation. In CVPR, 2022.
> > >
> > > [3] Making convolutional networks shift-invariant again. In ICML, 2019.
> > >
> > > [4] Delving deeper into anti-aliasing in Convnets. In IJCV, 2020.
> > >
> > > [5] Frequency LowCut Pooling -- Plug & Play against Catastrophic Overfitting. In ECCV, 2022.
> > >
> > > [6] Fix your downsampling ASAP! Be natively more robust via Aliasing and Spectral Artifact free Pooling. In Arxiv, 2023.
> > >
> > > [7] Real-Time Single Image and Video Super-Resolution Using an Efficient Sub-Pixel Convolutional Neural Network. In CVPR, 2016.
> > >
> > > [8] Task-Aware Image Downscaling. In ECCV, 2018.
> > >
> > > [9] Learning strides in convolutional neural networks. In ICLR, 2022.
> > >
> > > [10] Learning to Zoom and Unzoom. In CVPR, 2023.
> > >
> > > [11] Efficient segmentation: Learning downsampling near semantic boundaries. In CVPR, 2019.
> > >
> > > [12] SALISA: Saliency-Based Input Sampling for Efficient Video Object Detection. In ECCV, 2022.
> > >
> > > 2.Yes. It indicates Sec. 2 Stride Extension of supplementary material, which discusses the implementation of FouriDown at other strides based on extended theory as well as the diagram.

---

> > > > ### Author Response · Authors · 2023-08-19
> > > >
> > > > Dear Reviewer,
> > > > I truly appreciate your expertise and the valuable time you are investing in assessing our work. Please let me know if any additional information or clarification is required from our end. If your concerns have been addressed, we kindly hope that you consider revising your rating.

---

> > > > > ### Comment · Reviewer_k1g9 · 2023-08-21
> > > > >
> > > > > Thank you for careful response. I think the additional tables, figures, and description made this paper more comprehensive to be published at NeurIPS 2023. Therefore, I would like to recommend this paper to be accepted, and will change my rating.

---

### Official Review · Reviewer_fgKa · 2023-07-09

**Soundness:** 3 good
**Presentation:** 3 good
**Contribution:** 3 good
**Rating:** 6
**Confidence:** 4

**Summary:**

 - This paper explores a new method for downsampling, by factoring it into shuffling and superposing. Firstly, the authors provide an understanding of the aliasing in deep neural networks from a spectrum view.

 - FouriDown, a unified downsampling approach with the learnable and context-adaptive down-sampling operator, is proposed. This method is made of four key components: 2D discrete Fourier transform, context shuffling rules, Fourier weighting-adaptively superposing rules, and 2D inverse Fourier transform.

 - Extensive experiments on image de-blurring and low-light image enhancement are conducted, which consistently show that FouriDown can provide significant performance improvements.

**Strengths:**

 - Extensive qualitative and quantitative evaluation results on scale-sensitive tasks are conducted, which show the advantage of the proposed downsampling method over previous ones.

 - This work first provides an exploration of the aliasing issue in neural networks from a spectrum perspective, which will facilitate a deeper understanding of the aliasing issue.


**Weaknesses:**

 - Additional details, such as the computation costs and the respective results on images and features, should be provided. (See Question)

**Questions:**

 - What are the additional computation costs of FouriDown, compared to other downsampling methods?

 - Is there any difference between applying FouriDown to the image and feature level? It's interesting to see the comparisons with other downsampling methods in-between, as images and features in DNN have differences in frequency distribution.

**Limitations:**

Yes.

---

> ### Author Rebuttal · Authors · 2023-08-10
>
> 1. **FouriDown Computation**:
> Thank you for your question. The FouriDown incorporates the Fourier transform and its inverse, which traditionally have a computational complexity of \(O(N \log N)\).However, this part has been accelerated by pytorch library and does not occupy any FLOPs. Moreover, there are additional computations related to the spectral-shuffle operation and filtering in the frequency domain. For spectral-shuffle, the operations such as tensor remodeling and composition do not require additional Flops and the number of parameters. Moreover, the additional convolution layers contain very small flops and parameters. Finally, the computational cost of our algorithm compared to other downsampling methods is shown in the Table above. See attached file for more experiments.
> Note that although the shuffle process doesn't consume any parameters or FLOPs, implementing this step in software does take extra time. We will accelerate this step by CUDA and release the code in the future. We will further elucidate these computational trade-offs in the revised manuscript.
>
> 2. Thank you for raising this insightful question. FouriDown is a downsampling operator applied to the feature layer that relies on convolutional parameters. The learning downsampling process is similar to pixel-unshuffling. As a result, it may not be suitable for image domain resizing, such as bicubic and bilinear. However, it is meaningful to compare the feature map visualizations and their spectrum of different downsampling methods. Please refer to Figure 1-3 in the attached file. It can be seen that the spectrum following FouriDown obtains the outstanding smooth response in high and low frequencies.

---

### Official Review · Reviewer_e5Db · 2023-07-10

**Soundness:** 3 good
**Presentation:** 2 fair
**Contribution:** 1 poor
**Rating:** 2
**Confidence:** 4

**Summary:**

This paper presents a downsampling method for neural networks in the frequency domain.
The proposed downsampling consists of discrete Fourier transform, shuffling, superposing, and inverse discrete Fourier transform.
The shuffling shuffles the patches in Fourier domain and the superposing adds the patches with learned and adaptive weights.
Experimental results present that the proposed method outperforms Gaussian filtering and the ideal low-pass filtering in frequency domain for neural networks on various computer vision tasks.


**Strengths:**

Downsampling in frequency domain is an interesting topic.
The proposed method and the experimental results are reasonable.
Performance improvements on enhancement tasks are impressive.

**Weaknesses:**

Experiments lack comparisons and analyzes to convince the effectiveness of the proposed method.

The proposed method uses additional convolution layers than compared methods, so the computational overheads (FLOPs and latency) should be compared.
It is worth noting that DFT and IDFT also require additional overheads compared to the original networks.

Comparisons to numerous image downscaling methods (bilinear, bicubic, etc) are missing.
In particular, pixel-downshuffling [A] with convolution performs a similar operation to the proposed method without Fourier transform.

Feature map visualization in frequency domain is missing.
Feature maps of convolution layers usually contain high frequency edges, thus it is curious how the proposed method works in the feature space.

Ablation study of additional convolution layers is missing.
Simple operations instead of the convolution layers such as max or mean will be an interesting ablation study.

[A] Channel Attention Is All You Need for Video Frame Interpolation, AAAI 2020


**Questions:**

Please see the weaknesses.

**Limitations:**

Please see the weaknesses.

---

> ### Author Rebuttal · Authors · 2023-08-10
>
> Firstly, we appreciate your acknowledgment of the topic and methodological design of our work. Complete and meaningful of the comparisons and analysis has been complemented by visualizations and tables in the rebuttal phase. If this paper is accepted, we commit to meticulously revising and organizing the experimental and analysis sections in the revision.
> 1.	** DFT overheads **:
> First, following [1,2], we perform DFT using \textit{torch.fft.fft2(x)} with few extra overhead. To further illustrate this point, we conduct a toy experiment shown in the following table. It could be seen that although the FFT and its inverse traditionally have a computational complexity of O(N log N), this part has been accelerated by pytorch library and does not occupy any FLOPs. Therefore, the consumption of time is also very little
>
> | SID          |   | FLOPs(G) | Time(s)  |
> |--------------|---|----------|----------|
> | wo FFT/IFFT  |   | 13.753   | 0.0134   |
> | w FFT/IFFT   |   | 13.753   | 0.0139   |
> | w 4*FFT/iFFT |   | 13.753   | 0.0146   |
>
> [1] Intriguing Findings of Frequency Selection for Image Deblurring. In AAAI, 2023.
> [2] Deep Fourier-Based Exposure Correction Network with Spatial-Frequency Interaction. In ECCV, 2022.
>
> 2.Comparison with Traditional Downscaling Methods. Due to space limitation, in the supplementary, we recognize the importance of comparing our method with traditional techniques like nearest/bicubic/bilinear/lanczos downsampling shown in Fig 5. As depicted in Supp. Line 86-90, it can be observed that when the down-sampling factor is exactly divisible by the original image resolution, no interpolation actually takes place. In such cases, the bicubic/bilinear/lanczos methods mentioned above are equivalent to the nearest neighbor method, which appears same static weights [0.25,0.25,0.25,0.25] in FouriDown.
> Furthermore, we include comprehensive comparisons with these methods as shown in the above table.
>
> 2.Pixel-shuffling and proposed spectral-shuffling:  We are aware of the pixel-downshuffling method you referred to in [A]. However, it is hard to agree with your opinion. Specifically, there are following distinct differences.
>
> (a)Domain of Operation. Our method operates in the frequency domain, leveraging the unique properties of Fourier transforms. Pixel-downshuffling operates in the spatial domain, and the nature of these domains and their inherent properties can lead to different results.
>
> (b)Distinct Shuffle Patterns. It is apparent that our shuffle mode is specifically designed for spectrum and is completely different from the pixel-shuffle[A] mode which relies on nearby sampling. To illustrate the point better, we replace our spectrum-shuffle with pixel-shuffle after FFT as evidenced in following table, which leads to bad results.
>
> | | | |  | |
> |--------|--------------------|---|-------|---------|
> | MPRNET | FouriDown  |   | PSNR  | SSIM    |
> |  | wo shuffle         |   | 25.43 | 0.8144  |
> |   | w pixel-shuffle    |   | 23.43 | 0.7954  |
> |  | w spectral-shuffle |   | 30.31 | 0.8996  |
>
>  (c)White-box vs. Black-box Approach. Different from the pixel-downshuffle[A] operating as a 'black box', our shuffle criterion is derived from rigorous signal theory, ensuring a more interpretability and expansibility grounded process.
> (d)Potential Impact for FFT Developing. We believe the spectrum-shuffle isn't merely a technical novelty, it has significant implications for the develop of the Fourier theory in the low-level vision task.
> Above all, the Fourier domain offers insights and opportunities that are not readily available in the spatial domain. By working in this domain, our method provides a novel solution for adaptive aliasing issue and potential improvements in downsampling.
>
> 3.Feature map visualization. Thank you for your suggestions. We have included visualizations of the feature maps and their corresponding spectra in the PDF file attached to the rebuttal. Notably, the features extracted by FouriDown demonstrate a significantly larger response than other methods, attributing this to the unique global modeling mechanism we employ in the frequency domain. Moreover, to delve deeper into the reasons for its efficacy, we have also compared the spectral images of the features. The results show that the spectrum using FouriDown exhibits an exceptionally smooth response across both high and low frequencies.
>
> 4.Max or Mean Ablation Study. Indeed, this is an interesting experiment which highlights the extensibility of the FouriDown framework. In fact, we have already conducted this ablation study earlier, but due to space constraints, it was not included. As shown in the following table, according to Theorem 2, when averaging is used, FouriDown essentially becomes equivalent to the stride=2 method. The Max approach is indeed an interesting idea, however, regrettably, it hasn't been so effective. We hope to find the scenarios where the Max method is more applicable in the future.
> |            |        |   |       |         |
> |------------|------------|---|-------|---------|
> | DeepDeblur | FouriDown  |   | PSNR  | SSIM    |
> |            | Max        |   | 29.32 | 0.8815  |
> |            | Average    |   | 29.34 | 0.8817  |
> |            | Conv layer |   | 29.44 | 0.8856  |

---

> > ### Comment · Senior_Area_Chairs · 2023-08-19
> > **Discussion**
> >
> > Dear e5Db,
> > Thanks for your review! Could you please indicate that you have read the rebuttal and state whether or not your concerns are addressed?
> >
> > Thanks and best,
> > SAC

---

> > > ### Comment · Reviewer_e5Db · 2023-08-19
> > >
> > > Thanks for the response. It partially alleviates my concerns.
> > >
> > > However, after carefully reading the rebuttal, the contributions of this paper seem to be overclaimed.
> > >
> > > First, the work in [B] previously explores the aliasing problem in deep neural networks.
> > > The author's claims are not convincing given that this paper does not provide enough references for deep learning methods in frequency domains such as [C, D].
> > >
> > > Second, the proposed method just slightly outperforms compared methods in many tasks (deblurring, denoising, and dehazing), while outperforming them significantly in image/UHD enhancement.
> > > The manuscript does not address the performance gap across tasks.
> > >
> > > The reviewer suggests focusing the tasks on image/UHD enhancement and analyzing the performance improvements through feature visualization and comparisons for the next conference.
> > > The current manuscript does not address a basic question, 'What is the relation between image enhancement and the aliasing problem?'.
> > >
> > >
> > > [B] Revisiting Light Field Rendering with Deep Anti-Aliasing Neural Network, TPAMI 2022
> > >
> > > [C] Fourier Features Let Networks Learn High Frequency Functions in Low Dimensional Domains, NeurIPS 2020
> > >
> > > [D] FCNN: Fourier Convolutional Neural Networks, ECML PKDD 2017

---

> > > > ### Author Response · Authors · 2023-08-20
> > > >
> > > > 1.Thank you for your response. For the first problem, although the provided references have some relevance to our work, I believe they are not essential.
> > > >
> > > > (1) Reference [B] addresses aliasing using the Gaussian kernel. However, in the related work section, many similar works [8,10,11,12,13] as [B] have already cited. So, it is difficult to understand the necessity except for only increasing the number of references.
> > > >
> > > > (2) Our work is entirely different from [B]. Instead of addressing aliasing, we introduce an adaptive frequency superposition down-sampling paradigm, which expands the solution space of previous down-sampling methods in the deep learning.
> > > >
> > > > (3) There might be some misunderstandings regarding our contribution. The focus of our work is on the down-sampling operator, and the spectrum merely serves as an effective tool in the design of FouriDown. Hence, I believe it is not very necessary to introduce additional deep learning methods [C,D] in the frequency domain.
> > > >
> > > >
> > > >
> > > >
> > > > 2.Due to the varying datasets and degrees of difficulty among different tasks, it's quite common for a generic operator to yield different performance improvements across these tasks. For instance, even widely-used down-sampling methods such as max-pooling, pixel-unshuffle, and bicubic cannot guarantee consistent performance gains across these tasks.
> > > >
> > > > Moreover, evaluating the effectiveness of the proposed module in different tasks based on the absolute differences in PSNR and SSIM is not reasonable. This is because benchmarks derived from different scenarios and varying degradations inherently have significant variations. Thus, the experiments conducted in the manuscript across different tasks and network structures sufficiently demonstrate the good generalizability of our method in low-level vision.
> > > >
> > > > 3.In the revised version, we will carefully consider your suggestions to prioritize feature visualization and analysis on image enhancement to make our work more comprehensive and convincing.
> > > >
> > > > 4.For the last problem, there might still be some misunderstandings.
> > > >
> > > > The aliasing usually denotes static frequency superposition. Although it inspires our work, our main attention is not on addressing aliasing. Instead, we emphasize adaptive interaction between high and low frequencies during down-sampling.
> > > > After resolving misunderstandings, the relation of our work to the image restoration is as follows.
> > > >
> > > > (1) It's widely recognized that high and low frequencies play essential roles in image restoration and they correspond to the color and texture in the spatial domain of images, respectively. Therefore, by achieving adaptive the superposition of high and low frequencies during down-sampling, we facilitate restoration networks to better perceive degraded areas and restore them.
> > > >
> > > > (2) In the total comment, we have further emphasized and proved this point.

---

### Author Rebuttal · Authors · 2023-08-10

Thank all reviewers for insightful comment. We recognize the importance of providing comprehensive benchmarks for our proposed method. As shown in table below, we include results from traditional downsampling techniques like bicubic, bilinear, pixel-unshuffle, 2x2 learnable CNN (with stride=2), max-pooling, average-pooling, LPF, Gaussian and Ours. Noting that the “Original” downsampling of the method is pointed by asterisk (‘*’). This will allow a clearer contrast and showcase the advantages of our method not only against anti-aliasing approaches but also against these conventional downsampling methods.
Note that although the shuffle process doesn't consume any parameters or FLOPs, implementing this step in software does take extra time. We will accelerate this step by CUDA and release the code in the future. We will further elucidate these computational trade-offs in the revised manuscript.

| Method | Config          |   | LOL    |        | FLOPs(G) | Time(s) | Para(M)  |
|--------|-----------------|---|--------|--------|----------|---------|----------|
|        |                 |   | PSNR   | SSIM   |          |         |          |
| SID    | Bicubic         |   | 21.35  | 0.8497 | 13.764   | 0.0131  | 7.84     |
|        | Bilinear        |   | 21.26  | 0.8464 | 13.764   | 0.0136  | 7.84     |
|        | Pixle-shuffle   |   | 21.41  | 0.8552 | 13.954   | 0.0138  | 8.11     |
|        | Stride Conv     |   | 21.36  | 0.8534 | 13.954   | 0.0144  | 8.11     |
|        | Max pooling *   |   | 21.46  | 0.8584 | 13.753   | 0.0134  | 7.84     |
|        | Average pooling |   | 21.34  | 0.8481 | 13.754   | 0.0128  | 7.84     |
|        | LPF             |   | 21.79  | 0.8612 | 16.102   | 0.0149  | 8.54     |
|        | Gaussian        |   | 20.74  | 0.8124 | 16.102   | 0.0137  | 8.54     |
|        | Ours            |   | 23.28  | 0.8708 | 13.827   | 0.0176  | 7.87     |

| Method | Config          |   | DVD     |        | FLOPs(T) | Time(s) | Para(M)  |
|--------|-----------------|---|---------|--------|----------|---------|----------|
|        |                 |   | PSNR    | SSIM   |          |         |          |
| MPRNET | Bicubic         |   | 29.8302 | 0.8815 | 1.398    | 0.5258  | 15.74    |
|        | Bilinear        |   | 29.8795 | 0.8822 | 1.398    | 0.5422  | 15.74    |
|        | Pixle-shuffle   |   | 29.8202 | 0.8816 | 1.399    | 0.575   | 15.93    |
|        | Stride Conv *   |   | 30.12   | 0.8958 | 1.399    | 0.5082  | 15.93    |
|        | Max pooling     |   | 29.8038 | 0.881  | 1.398    | 0.5402  | 15.74    |
|        | Average pooling |   | 29.8716 | 0.8825 | 1.398    | 0.5414  | 15.74    |
|        | LPF             |   | 30      | 0.8918 | 1.416    | 0.5004  | 16.26    |
|        | Gaussian        |   | 30.23   | 0.8922 | 1.416    | 0.5491  | 16.26    |
|        | Ours            |   | 30.31   | 0.8996 | 1.398    | 0.5974  | 15.93    |

---

> ### Comment · Reviewer_k1g9 · 2023-08-14
>
> [Regarding the attached file]
>
> In the Figures 1 and 3, the authors claim that a larger response of FouriDown is achieved by global modeling. However, it seems that a larger response itself cannot equal to better performance because Gaussian filter also obviously shows large response but performs poorly on low-light enhancement when compared to the others except FouriDown. Moreover, responses of the others are similar but performances differ each other. What is the significance of response value in feature map level? How could this feature map visualization support the performance gains?
>
> In the contrary, Figure 2 conveys the effectiveness of FouriDown well. However, I believe that providing more detailed explanations (e.g., importance or meaning of smooth response in low and high frequencies) of this figure would enhance the comprehensiveness of this paper and incline this reviewer toward acceptance.
>
> The additionally provided tables are really thorough, which is encouraged to be included in the revised version.

---

> > ### Author Response · Authors · 2023-08-16
> >
> > 1.Thanks for your meticulous review. I apologize for the oversight; the data for LPF and Gaussian in the aforementioned table were inverted. The performance of these two methods should be consistent with the Table 1 of the submitted manuscript. We have made the necessary corrections below.
> >
> > | Method | Config   |   | LOL    | |
> > |--------|----------|---|--------|--------|
> > | |  |   | PSNR | SSIM |
> > | SID  | LPF |   | 20.74  | 0.8124 |
> > | | Gaussian |  | 21.79  | 0.8612 |
> >
> > Based on the revised table, the visualization results in Figure 1 and 3 above can reasonably be explained as following.
> >
> > (1) It can be observed that the model equipped with FouriDown generates much stronger responses to degradation-aware regions, i.e. global low-illumination in the low-light enhancement task. In contrast, the model with other down-sampling method responds weakly to these regions. The results demonstrates the effectiveness of FouriDown in capturing degradation-aware information by adaptive frequency superposition in down-sampling.
> >
> > (2) For the Gaussian method, as can be seen from Figure 1 and 3, its response to degradation is relatively large (second only to FouriDown), thus achieving performance that is also second only to FouriDown. Similarly, as the LFP method has the poorest performance, its feature response of the low-light areas is also the lowest. The performance of other methods is roughly similar, so their feature responses are also quite similar, indicating a similar capability to capture image degradation areas.
> >
> > (3) Additionally, from the spectral comparison in Figure 2, it can be observed that the Gaussian method loses a lot of high-frequency information compared to FouriDown. This leads to challenges in recovering textures and details in dark areas. Hence, although the Gaussian method exhibits good responses, FouriDown achieves better performances compared to it.
> >
> > 2.Thank you for acknowledging the effectiveness of our Figure 2. Due to space constraints, we provided a brief analysis of Figure 2. Now, we will delve deeper into the discussion.
> >
> > As illustrated in Figure 2, FouriDown achieves a very distinctive feature frequency response compared to other methods. I believe this spectral distribution pattern offers the following advantages:
> >
> > (1) **High Performance**: Compared to other methods, our FouriDown adaptively adjusts the high and low frequencies, resulting in a wider-band response in the output feature spectrum (as observed in the four corners of Figure 2(j)). Contrasted with previous methods that used fixed frequency aliasing patterns, our approach activates a broader bandwidth on the spectrum, bringing the enhanced performance in image restoration.
> >
> > (2) **Good Generalization**: Different scenes possess unique spectral characteristics. For instance, the spectrum of the input image in Figure 2 displays larger response values at the intersections with the zero frequency line, indicating that this scene contains more frequency components in these specific directions. As shown in Figure 2, previous downsampling methods tend to preserve or amplify such scene-specific frequency bands. In contrast, our FouriDown attenuates these scene-specific bands via modulating the interaction between high and low frequencies, which reduces the dependence on the scene. Consequently, as demonstrated in the following table, FouriDown achieves superior generalization performance in Huawei dataset compared to other methods.
> >
> > | Config          |   | LOL --> Huawei |        |
> > |-----------------|---|----------------|--------|
> > |    |   | PSNR           | SSIM   |
> > | Bicubic   |   | 17.2816        | 0.6219 |
> > | Bilinear     |   | 17.2546        | 0.6206 |
> > | Pixle-shuffle   |   | 17.6263        | 0.6272 |
> > | Stride Conv     |   | 17.1293        | 0.6187 |
> > | Max pooling *   |   | 17.2724        | 0.6205 |
> > | Average pooling |   | 17.8991        | 0.6250  |
> > | LPF  |   | 16.9921        | 0.5862 |
> > | Gaussian        |   | 17.1723        | 0.6132 |
> > | Ours   |   | 19.4855        | 0.6569 |
> >
> > (3) **Less artifacts**: The existing downsampling, due to sub-optimal interactions between high and low frequencies, may lead to a proliferation of artifacts in restored results. As shown in Figure 2, most of the downsampling methods' output spectrums display numerous interruptions between frequency bands. Moreover, some even manifest potentially detrimental ring-shaped frequency bands, e.g. in max-pooling. Such spectral distributions diverge from the typical patterns of natural images, potentially resulting in undesired artifacts in the results. In contrast, our method ensures a smoother transition during the activation of high and low frequencies, leading to the restoration with significantly fewer artifacts. More qualitative comparisons will be provided in the revised version.
> >
> > 3.Thank you for your constructive feedback on the additional tables and figures. We will add the above table and visualization analysis to the revised version.

---

> > > ### Comment · Reviewer_k1g9 · 2023-08-16
> > >
> > > 1. The corrected table shows consistency with Figures 1 and 3.
> > >
> > > 2. The results of Figure 2 also becomes more comprehensive. I think this part must be included in the final paper, at either main content or supplementary material. Moreover, it is encouraged to include more detailed explanation of the FFT magnitude images of Figure 2 for those who are not familiar with frequency domain, such as the mean of colors, four corners, or cross lines in each figure. This explanation does not have to be posted on this official comment, but to be presented in the paper.
> > >
> > > 3. Thank you for response.

---

> > > > ### Author Response · Authors · 2023-08-17
> > > >
> > > > Thank you very much for your kind responses and suggestions. We will definitely include the aforementioned comprehensive comparisons and analyses in the final paper to make it more comprehensive and convincing.

---

### Decision · Program_Chairs · 2023-09-21

**Decision:**

Accept (poster)

**Comment:**

This submission has received five reviews with four reviews that recommend acceptance and one reviewer with a strong rejection vote.

All reviewers appreciate the problem, this is relevant as a basic building block for network architectures. The experiments are thorough and comprehensive, several image processing tasks with several baselines each, the results show an increased performance throughout. This includes image classification and image processing experiments, on the classification part only small models and CIFAR datasets.


We checked carefully the concerns of reviewer e5DbChecking carefully the raised concerns of reviewer e5Db. However after the rebuttal and the discussion we find that the concerns are mainly answered, eg. the experimental evidence is found, the rebuttal states computation time and other baseline results were already in the initial draft. The related work that was raised in the discussion is not a critical miss for this paper. Therefore we regard the strong reject rating as too drastic given the arguments.

There are certainly points that can and should be improved, the rebuttal and discussion surfaced some. Also many typos need correct, e.g.  (Guassian, Groud truth) also maybe move proof to the appendix/supplementary? However the results are convincing, the method is found novel from all reviewers, therefore the recommendation is to accept the work.